# Central Causation of Autism/ASDs via Excessive [Ca^2+^]i Impacting Six Mechanisms Controlling Synaptogenesis during the Perinatal Period: The Role of Electromagnetic Fields and Chemicals and the NO/ONOO(-) Cycle, as Well as Specific Mutations

**DOI:** 10.3390/brainsci14050454

**Published:** 2024-04-30

**Authors:** Martin L. Pall

**Affiliations:** School of Molecular Biosciences, Washington State University, Pullman, WA 99164, USA; martin_pall@wsu.edu

**Keywords:** calcium-regulated protein kinases, ASD causation via high and low [Ca^2+^]i, EMF and chemical [Ca^2+^]i elevation, NO/ONOO(-) elevation in ASDs, therapeutic Nrf2 elevation, electric currents and EMFs

## Abstract

The roles of perinatal development, intracellular calcium [Ca^2+^]i, and synaptogenesis disruption are not novel in the autism/ASD literature. The focus on six mechanisms controlling synaptogenesis, each regulated by [Ca^2+^]i, and each aberrant in ASDs is novel. The model presented here predicts that autism epidemic causation involves central roles of both electromagnetic fields (EMFs) and chemicals. EMFs act via voltage-gated calcium channel (VGCC) activation and [Ca^2+^]i elevation. A total of 15 autism-implicated chemical classes each act to produce [Ca^2+^]i elevation, 12 acting via NMDA receptor activation, and three acting via other mechanisms. The chronic nature of ASDs is explained via NO/ONOO(-) vicious cycle elevation and MeCP2 epigenetic dysfunction. Genetic causation often also involves [Ca^2+^]i elevation or other impacts on synaptogenesis. The literature examining each of these steps is systematically examined and found to be consistent with predictions. Approaches that may be sed for ASD prevention or treatment are discussed in connection with this special issue: *The current situation and prospects for children with ASDs*. Such approaches include EMF, chemical avoidance, and using nutrients and other agents to raise the levels of Nrf2. An enriched environment, vitamin D, magnesium, and omega-3s in fish oil may also be helpful.

## 1. Introduction

Autism is a childhood disorder characterized by early expressed impairments in social interaction and communication, and repetitive or circumscribed interests or behavior [1,2,3,4]. Autism spectrum disorders (ASDs) include autism, pervasive developmental disorder, and Asperger’s syndrome, and are thought to be a true spectrum without discrete divisions between these three types. The goal here is not to focus on the behavioral aspects of autism/ASDs, but rather to propose a causal/explanatory model which explains the neurodevelopmental, physiological, biochemical, environmental, and genetic causal mechanisms of ASDs. Those, in turn, suggest both therapeutic and preventative approaches to ASDs. The complexity of the presented model requires the presentation of the model first. Diverse experimental and epidemiological findings follow throughout most of this review, testing each individual mechanism in the proposed mechanism. 

Figure 1 is an outline of the mechanisms proposed here and, in some cases, elsewhere, to act during the perinatal period of brain development [5], to disrupt synaptogenesis and therefore cause autism spectrum disorders (ASDs). Accordingly, key factors in autism/ASD causation are proposed to be the six mechanisms in the lower left part of Figure 1 (in red), each of which have been shown to have important roles in synaptogenesis in the developing brain. These six mechanisms are neuronal migration, dendritic outgrowth, synapse formation, synapse maturation, synaptic pruning, and lowered MeCP2 function. It is shown below that of these six mechanisms are regulated by intracellular calcium ([Ca^2+^]i). 

This model predicts that stressors which may act to cause the autism/ASD epidemic [4,6,7] must act either directly or indirectly to raise [Ca^2+^]i levels. The issue of whether the autism/ASD epidemic is real has been widely debated in the scientific literature. Each of the three reviews cited here [4,6,7] come to the conclusion that it is. I find the most convincing of these to be the Nevison et al. [4] review, which estimates an autism incidence going from 0.001% in 1931 to 1.2% in 2014, and concludes that the very large, circa 1200-fold, apparent increase cannot be due to diagnostic substitution. I find the arguments presented in Nevison et al. [4] to be convincing. I also know that my opinion has been challenged by others. I will come back to this issue twice later in the paper. 

While there may be many possible stressors that may cause the large apparent increase in the incidence of autism/ASDs, there are two main classes of stressors where exposure increases in recent years may provide an explanation for the ASD epidemic, which are electromagnetic fields (EMFs) and diverse chemical exposures. Electromagnetic fields (EMFs), as is documented below, act primarily via activation of voltage-gated calcium channels (VGCCs) to consequently raise [Ca^2+^]i levels and diverse chemicals, which often act primarily via activation of the NMDA receptors (NMDA-Rs), also raise [Ca^2+^]i levels (see blue parts of Figure 1). 

While synaptogenesis and pruning continue throughout our lifespans, so that it may be possible to reverse much of the dysfunctional synaptogenesis in autism/ASDs, there are two important mechanisms which are proposed to produce much of the chronic nature of autism (Figure 1). One of these is the NO/ONOO(-) cycle (discussed in Section 6.1), a vicious cycle which is produced by mechanisms that act at the level of individual cells, which can explain important properties of autism, including the roles of oxidative stress, inflammation, and mitochondrial dysfunction. A second mechanism producing chronic effects is the MeCP2 mechanism, which produces calcium-regulated DNA methylation and therefore, has apparent epigenetic effects. While both the NO/ONOO(-) cycle and MeCP2 function each produce challenges for effective autism treatment, this does not mean that ASDs are untreatable. One approach to treatment may be to raise the levels of Nrf2 (Figure 1), a transcription factor that raises many cytoprotective responses and is predicted to lower the NO/ONOO(-) cycle (discussed in Section 6.1).

Finally, on the left side of Figure 1, syndromic and non-syndromic forms of autism/ASDs are produced by specific mutations, which may be either familial or newly occurring (de novo) mutations. Such mutations are typically of five different types. Some involve mutations of genes encoding proteins which have direct roles in synapse development and/or function, or genes with regulatory roles which may directly or indirectly produce increases in [Ca^2+^]i levels (see Section 1.1, Section 2.2, Section 3.1 and Section 5).

The reader should, of course, be skeptical about each proposed mechanism in Figure 1 until sufficient evidence is provided documenting such mechanism. Healthy skepticism is the basis of all good science. 

Much of this paper provides extensive evidence for the specific mechanisms claimed here and their roles in autism causation.

### 1.1. Autism/ASD Causation via Disruption of Synaptogenesis during the Perinatal Period: A Brief Review of Much of the Evidence

It is widely reported that there is very rapid development of synapses during the perinatal period. Such synaptogenesis is dysfunctional in both human autism/ASD patients and in autism/ASD animal models [8,9,10,11,12,13,14,15,16,17,18,19,20,21,22,23,24,25,26]. Furthermore, synaptogenesis dysfunction is raised by agents causing autism [10,12,14,24,26] and lowered by agents thought to be helpful in autism treatment [18,19,23]. Synaptogenesis dysfunction may be a major cause of the changes in connectivity between different regions of the brain in ASDs [27,28,29,30,31]. Furthermore, many of the behavioral changes that occur in autism have been proposed to be caused by changes in brain region connectivity [27,29,32,33,34,35]. 

Let us examine Figure 1, which outlines what is argued here, to be the central set of mechanisms in autism. The core of Figure 1 is that there are six mechanisms which regulate synaptogenesis, with each of these being regulated by intracellular calcium ([Ca^2+^]i). Those six are neuronal migration, dendritic outgrowth, synapse formation, synapse maturation, synaptic pruning, and MeCP2 function. Mutational inactivation of MeCP2 is the central cause of Rett syndrome, which is considered to be an ASD. MeCP2 is a calcium-inhibited DNA methylation enzyme which has a general role in ASD, controlling DNA methylation and, therefore, epigenetic changes. 

Each of the six mechanisms that regulate synaptogenesis have been shown to be abnormal in autism/ASD patients: neuronal migration [36,37,38,39], dendritic outgrowth [10,40,41], synapse formation [8,10,15,42], synaptic maturation [8,42], synaptic pruning [8,15,43,44], and MeCP2 function [45,46,47,48]. Qiu and Cheng [48] linked changed MeCP2 function with increased ubiquitination and degradation of synaptic proteins, suggesting a plausible role of lowered MeCP2 function in producing increased synaptic pruning in ASDs.

I have claimed here that these six mechanisms that regulate synaptogenesis are each regulated by [Ca^2+^]i, but I have provided no evidence above of such calcium regulation except in the case of the calcium-inhibited MeCP2 enzyme function. Two important reviews, Krey and Dolmetsch 2007 [49] and Lohmann, 2009 [50], have each shown that [Ca^2+^]i signaling has a very important role in autism/ASD causation. These reviews demonstrated that five of the six mechanisms proposed here to have central roles in ASD causation are each aberrantly regulated by [Ca^2+^]i in ASDs (all except MeCP2 function). Lozac, 2010 [51] documented the role of not just [Ca^2+^]i elevation, but also voltage-gated calcium channels (VGCCs) in the early development of autism, which may suggest a special role of EMFs because of their action in activating the VGCCs (Figure 1). 

There are many reviews that have emphasized the important role of calcium signaling in autism/ASDs, in addition to the Krey and Dolmetsch 2007 [49] and Lohmann, 2009 [50] reviews. Liao and Li 2020 [52] state that “calcium-signaling pathways may serve as scaffolds that unite genetic lesions into a consensus etiology of ASDs. Pourtavakoli and Ghafouri-Fard [53] and Nguyen et al. [54] have each reviewed studies implicating excessive calcium signaling in ASDs. A key type of evidence reviewed in [49,51,52,53,54] is the highly repeated studies showing that diverse mutations that each produce excessive [Ca^2+^]i cause ASDs. Such genetic evidence of excessive [Ca^2+^]i causation is extraordinarily strong, much stronger than almost any epidemiological evidence of causation. 

A possible important mechanism by which [Ca^2+^]i may cause autism is by greatly increasing the activity of the most active calcium-activated protein kinase in the brain, calcium/calmodulin protein kinase II (CamKII). Elevated CamKII activity increases the incidence of autism/ASDs [55,56]. Sudarov et al. [57] showed that CamKII controls both synaptogenesis and autism-like behavior in mice. CamKII also regulates the activity of each of the six mechanisms proposed here to have important roles in both synaptogenesis and ASD causation: neuronal migration [58,59], dendritic outgrowth [58,60,61], synapse formation [58,62,63,64], synapse maturation [64,65,66,67], synaptic pruning [58,64,66,67], and MeCP2 activity [68,69]. While the studies cited in this paragraph implicate CamKII in producing aberrant synaptogenesis in autism/ASDs, they do not imply that CamKII-dependent protein phosphorylation is the sole or predominant [Ca^2+^]i-dependent causal mechanism of either synaptogenesis or ASD causation. 

Another study that links autism/ASD causation with aberrant synaptogenesis focuses on the finding that macrocephaly is more common in autism/ASD individuals than in normal individuals [1,70,71,72,73,74,75,76,77]. Macrocephaly has been shown to be caused by aberrant synaptogenesis [72,73,74,75,76,77,78]. This connection between aberrant synaptogenesis and macrocephaly is not novel, as shown in each of the six citations [72,73,74,75,76,77] cited twice in the previous two sentences. In addition, Huang et al. [72] discusses the role of aberrant synaptogenesis in producing hyperconnectivity in a mouse model of macrocephaly/autism syndrome.

In summary: Perinatal synaptogenesis is dysfunctional in human ASDs and animal models of ASD;Synaptic developmental dysfunction is raised by agents causing autism and lowered by agents found to be helpful in autism treatment;[Ca^2+^]i signaling is very important in autism/ASDs;Each of the six mechanisms that have important roles in synaptogenesis are regulated by [Ca^2+^]i and can therefore be impacted by either inappropriate [Ca^2+^]i elevation, or possibly even inappropriate [Ca^2+^]i depression;Each of those six mechanisms are impacted in ASDs;Each of those same six mechanisms are regulated by CamKII protein kinase, the most active calcium-regulated protein kinase in the brain;Synaptogenesis dysfunction produces tissue connectivity changes in the autism/ASD brain which are thought, in turn, to produce many autism symptoms. It may act, in part, to disrupt and change neural network formation, a possible central causal mechanism of autism/ASDs;Aberrant synaptogenesis causes macrocephaly which has much higher prevalence in individuals with autism/ASDs.

Because numbers 4, 5, and 6 each have six different supportive findings, the summary list here provides a total of 23 different types of evidence supporting the central mechanism of Figure 1—that autism causation is centered on excessive perinatal [Ca^2+^]i impacting six mechanisms, each of which regulate synaptogenesis, with impacts that collectively cause ASDs.

I am returning briefly to the question raised in Section 1, of whether the autism epidemic is a genuine epidemic. We have here unequivocal genetic evidence of [Ca^2+^]i causation of autism/ASDs. We also have above in this section, extensive evidence that excessive [Ca^2+^]i produces the diverse changes, in both synaptogenesis and in specific mechanisms involved in synaptogenesis, seen in ASD patients and animal models. This paper, in Section 2 and Section 3, provides extensive evidence that both EMFs and diverse chemicals each act via increased [Ca^2+^]i, and are plausible causes of the autism/ASD epidemic. You should, of course, be skeptical about statements that you have not yet had the opportunity to examine the evidence for. Those combinations of findings lead to the conclusion that the autism epidemic is real and that it is caused, at least in part, by both EMF exposures and diverse chemical exposures. 

## 2. How Electronically-Generated EMFs Act via Two Distinct Mechanisms to Activate the VGCCs and Produce Excessive [Ca^2+^]i 

In Figure 1, it is also proposed that electronically-generated electromagnetic fields (EMFs) act via activation of the voltage-gated calcium channels to allow large amounts of calcium to flow into the cells of our bodies, producing excessive [Ca^2+^]i which acts, in turn to cause synaptogenesis dysfunction and consequently, ASDs, The purpose of this section is to outline the properties of such electronically-generated EMFs, how they activate the VGCCs and other voltage-gated ion channels and, in a broader sense, how EMF effects are produced in our bodies. Following that, the multiple types of evidence linking such EMFs to autism/ASD causation will be discussed.

Electronically-generated EMFs are generated by electric currents. Electric currents were first shown to induce EMFs in the space around them in a paper published by Michael Faraday in 1831 [79]. This process is the central focus of Chapter 7 in Purcell 1995 [80]. Such electronically-generated EMFs are coherent, being emitted in a particular vector direction, with a particular frequency, polarity, and phase [81]. Coherence causes these EMFs to place strong electric forces and time-varying magnetic forces on electric charges in the cells of our bodies. In contrast, most natural EMFs are incoherent, being made up of astronomical numbers of photons emitted in different vector directions, with different polarities and phases, and often with different frequencies. Such incoherent EMFs only produce miniscule electric or magnetic forces. The focus in understanding how EMFs can produce biological effects must, therefore, be on the health impacts produced by electronically-generated EMFs via these forces on electric charges in the cells of our bodies. The critical role of both coherence and electric forces was recognized by both Fröhlich [82] and Brizhik [83] in their modeling of how non-thermal EMF effects may be generated.

It was shown that in 24 different studies, VGCC channel blockers could block or greatly lower effects produced by diverse EMFs, with frequencies ranging from microwave/radiofrequencies through intermediate frequencies, extremely low frequencies, and even nanosecond EMF pulses, static electric fields, and magnetic fields in [84]. That frequency range has subsequently been extended to include millimeter wave frequencies used in some 5G systems [81]. The VGCC calcium channel blocker studies clearly show that these electronically-generated EMFs act via activation of the VGCCs, leading to a rapid influx of calcium ions into cells and consequent increases in [Ca^2+^]i. It should be noted that EMFs induced almost instantaneous elevation of [Ca^2+^]i, as shown by Pilla [85], and that diverse frequency EMFs induce increases in calcium signaling, as reviewed in Walleczek [86]. The findings summarized in this paragraph clearly show that EMF exposures can act via excessive [Ca^2+^]i and allow us to predict that they can, at least in principle, cause the types of [Ca^2+^]i-dependent effects proposed in Figure 1 to produce synaptic developmental dysfunction and therefore cause autism/ASDs. The evidence discussed in Section 2.1, Section 2.2, Section 2.3, Section 2.4, Section 2.5, Section 2.6 and Section 2.7 below test that prediction.

Patch clamp studies have shown that EMF-induced VGCC activation, in most cases, is produced by direct forces on the voltage sensor, rather than by plasma membrane depolarization [87]. While voltage-gated sodium, potassium, and chloride channels, which are each controlled by a similar voltage sensor, have also been shown to be activated by the EMFs [87], these other channels have only relatively minor roles in producing biological effects, with the exception of a few cases where the sodium channels have major roles. The structure of these plasma membrane channels, how the voltage sensor acts to control the opening of the channels, and how the ion specificity of the channels is determined, has been reviewed by Catterall [88]. It should be noted that although [88] is mainly focused on the voltage-gated sodium channels, the properties of the voltage sensors of the VGCCs and the other voltage-gated ion channels are similar to each other. The ion specificity of these channels is determined by the properties of the specificity filter of each channel, which determines which ions can transit through the channels into or out of cells [88]. 

As reviewed in Catterall [88], voltage-gated ion channels all have a four-domain structure in the plasma membrane, with four similar domains found at approximately 90 degrees from each other. Each domain has six alpha-helixes, designated S1 through S6, with each of the S4 helixes containing from four to seven positively charged groups on arginine or, in a few cases, lysine side chains. The four conserved charged groups in the S4 helixes are three residues apart. They act to ratchet the S4 helixes in a counter-clockwise direction, with the S4 helix charged side chains ratcheting against fixed negative charges, or hydrogen bonding side chains in the S3, S2, or S1 helixes. It is proposed that both electric and time-varying magnetic fields place forces directly on the S4 helix charges, and that peak forces are much more important than average forces, providing an explanation for the importance of pulsation in producing much higher biological effects.

The high-intensity nanosecond pulses are an exception to the EMFs acting directly via forces placed on the voltage sensor ratcheting mechanism. High intensity nanosecond pulses have been shown to act via electroporation and by consequent depolarization of the plasma membrane [89,90], such that depolarization activates these channels. Nanosecond pulses also produce biological effects, predominantly via VGCC activation, as shown by the VGCC calcium channel blocker studies [84,91].

VGCC activity acts via [Ca^2+^]i to produce diverse biological effects via the pathways shown in Figure 2.

There are two main pathophysiological pathways of action following electronically-generated EMF exposure (Figure 2), the excessive calcium signaling pathway and the peroxynitrite/free radical/oxidative stress/NF-kappa B/inflammation pathway. Both are involved in many, but not all pathophysiological EMF effects. In contrast, relatively modest increases in [Ca^2+^]i act via nitric oxide (NO) signaling, cGMP, and elevated Nrf2 to produce therapeutic effects. The therapeutic pathway and the peroxynitrite/oxidative stress/inflammation pathway produce opposite effects, with each of these two pathways inhibiting the other [91]. 

Before moving on, it is worth discussing one particularly important study where EMF exposures were shown to act via VGCC activation to produce many different effects. That is the El Swefy et al. [92] study, where 2 h per day of low-intensity 3G mobile phone base station radiation produced massive neurodegeneration in the brains of rats, with approximately 34% of the brain cells dying in 4 weeks. El Swefy et al. [92] showed that 11 measured effects and four observed behavioral effects were each largely blocked by amlodipine, one of the dihydropyridine VGCC blockers. These findings argue that there are diverse changes produced by pulsed EMF exposures in this study, which had roles in producing profound neurodegeneration in four weeks, that were each produced via VGCC activation. Furthermore, as discussed in [91], each of those 11 measured EMF-induced changes in [92] can be produced via the EMF effects outlined in the two pathophysiological pathways in Figure 2.

### 2.1. EMFs as a Possible Major Cause of the Autism/ASD Epidemic

A number of individual scientists and research groups have suggested that increased autism/ASD incidence is associated with, and may be caused by, electromagnetic field (EMF) exposure [5,93,94,95,96,97,98,99,100,101,102,103,104]. These are based on both correlations between increased ASD incidence and increased EMF exposures, and other types of evidence. There have been a series of technological changes that have increased EMF exposures over the years, including large increases in microwave ovens; cordless phones, satellite phones and pagers, largely between 1984 and 1997; cell/mobile phones and cell phone towers (also known as mobile phone base stations), starting in the late 1990s; huge increases in Wi-Fi systems, starting in the early 2000s; smart meters, and very large amounts of dirty electricity in power lines, starting circa 2005. Because pulse modulation is used in all wireless communication, except for FM radio and 2G cell phones, and such pulse modulation has been shown to make EMFs much more biologically active than non-pulsed EMFs of the same average intensities, as reviewed in [105,106,107,108,109], EMF pulse modulation is, therefore, of great concern. The development of ever “smarter”, more pulse-modulated devices is, therefore, of much greater concern.

How, then, does this paper propose that EMFs act to produce autism/ASDs? If one compares Figure 1 with Figure 2, one pathway of action should be obvious. EMF-induced [Ca^2+^]i produces excessive calcium signaling, producing changes in the activities of each of the critical mechanisms impacting synaptogenesis. When this occurs during the perinatal period, ASDs are the consequence. However, what may be less obvious is that the other main pathophysiological pathway, the elevated peroxynitrite/reactive free radical/oxidative stress/NF-kappaB elevation/inflammatory cytokine pathway, is entirely elevated in patients with autism/ASDs, as documented in Section 6, and therefore has an important role, as well.

### 2.2. Genetic Evidence for VGCC Activity Roles in Autism

Timothy syndrome is a rare devastating genetic disease whose features include [110] “autism and autism spectrum disorders along with cardiac arrhythmia and developmental abnormalities”. There are four types of L-type VGCCs and the one that is most active in the brain, heart, and many other tissues, Cav1.2, is mutated in Timothy syndrome [110,111,112]. That mutation causes a change in the pore forming subunit of the L-type VGCC, such that the closing of the channel that allows calcium to flow into the cell is dysfunctional [110,111,112]. In the wild type L-type VGCC channel, the calcium flow into the cell and plasma membrane, and partial depolarization, on activation of the channel, acts to close the channel, thus limiting calcium influx. In Timothy syndrome, the closing of the channel is greatly slowed, such that the L-type VGCCs encoded by the mutant gene are, in effect, greatly hyperactive [110,111,112]. It follows from this that excessive L-type VGCC activity can act in the developing fetus and/or after birth to cause autism and ASDs. It has therefore been suggested [110] that the Timothy syndrome mutation offers a “promising starting point for exploring the underlying pathophysiology of autism”. A transgenic mouse model of Timothy syndrome was found to show autism-like symptoms, [113] including “markedly restricted, repetitive, and perseverative behavior, altered social behavior, altered ultrasonic vocalization, and enhanced tone-cued and contextual memory following fear conditioning”.

Timothy syndrome mutations are just the beginning of the VGCC mutational effects. Many different mutations, in 13 different genes, impact VGCC activity, each producing ASD-like effects, as reviewed in [52,114]. While many such mutations produce the effects of ASDs via gain of function and increased VGCC activity, others produce such effects via loss of function and lowered VGCC activity. How loss of function mutations with lowered VGCC activity can cause autism/ASDs is discussed at the end of Section 4. These very extensive genetic findings clearly show that increased activity of multiple VGCCs does cause autism/ASDs in both humans and animal models. These gain of function findings are most relevant to the EMF mechanism of action. It is important to note that even modest genetically-caused increases in VGCC activity can cause increases in autism/ASD incidence [52,114]. These findings greatly strengthen the inference that even modest EMF exposures during the perinatal period may cause autism/ASDs. 

### 2.3. EMF Effects on Neurite Outgrowth from Neural Stem Cells

It has also been shown that low-intensity EMF exposures act on neural stem cells to inhibit neurite outgrowth [115,116,117,118,119,120]. Because such neurite outgrowth is an essential precursor process for synaptogenesis, such effects may be predicted to greatly lower synapse formation in the developing brain. Chen et al. [115] showed that EMFs acted to inhibit neurite outgrowth by inhibiting EPHA5 signaling. Akaneya et al. [121] have shown that EPHA5 also has a role in controlling synaptogenesis and consequently, EMFs are predicted to impact the process of synaptogenesis via lowered EPHA5 signaling. EPHA5 especially stimulates the development of synapses [121] that stimulate the NMDA receptors and consequently, as discussed below in Section 3.2, will therefore interact with [Ca^2+^]i signaling. Because Abdul-Wajid et al. [122] have shown that the T-type VGCC activity inhibits EPHA5 expression, these findings confirm how EMFs can act via VGCC activation to inhibit neurite outgrowth and synaptogenesis of neural stem cells. Kaplan et al. [123] suggested that EMFs inhibit the development of neural stem cells. 

In conclusion, EMFs inhibit neurite outgrowth in neural stem cells, a process that is essential for synaptogenesis. Because there are many more neural stem cells in the brain during the perinatal period than there are after that period, this may be an additional reason why the perinatal period may have a key role in the development of ASDs.

### 2.4. Epidemiology of EMFs and Autism/ASD Prevalence

Bertrand et al. [124] showed that Brick Township, a town on the New Jersey coast of the U.S., where an important military radar station is located, had an elevated prevalence of autism, suggesting but not proving the radar-associated EMF causation of autism. The Bertrand et al. [124] paper has been cited 995 times at the time of writing (Google Scholar), so the importance of this paper has been widely recognized in the scientific literature. Although the [124] paper was not designed to suggest at what stage(s) in development EMF exposure may be acting to cause autism, the Klinghardt study [96] does. Klinghardt [96] studied the EMF exposure levels of pregnant women who gave birth to babies with autism compared with normal control babies. The study [96] found that the EMF exposure levels for autism babies were greatly elevated by both where the mother slept during pregnancy, and where the baby slept in the months following birth, compared with control pregnancies producing normal babies. Roelfsema et al. [125] studied the apparent role of school EMFs in autism. This suggests that the developmental stage where EMFs can influence the development of autism may go well past the perinatal period, into school age children.

The association of prenatal or perinatal EMF exposure with speech problems in children has also been reported. Zarei et al. [126] found a statistically significant association of speech problems with maternal mobile phone usage. In a larger, more recent study, Zarei et al. [127] found that speech problems had a statistically significant association with both maternal cordless phone usage and living near high voltage power lines, two additional important sources of EMF exposure. In the more recent [127] study, maternal mobile phone usage did not have a statistically significant association with speech problems, a result that was attributed by the authors as being due to very few mothers not using mobile phones. 

I already knew about [96,124] before starting this paper and found [123,124,125] by searching “autism” in the EMF-Portal database. Having failed to find any further epidemiological studies on EMFs and autism, I believe these five may be all that exist. Each of the five human epidemiological studies cited here provide evidence in support of the view that EMF exposure may have a role in causing the autism epidemic. What is surprising, however, is that despite widespread concern about a possible causal link between EMF and autism/ASDs [5,93,94,95,96,97,98,99,100,101,102,103,104] and the widespread concern about the autism/ASD epidemic, there have been few human epidemiological studies. 

### 2.5. EMF Causation of ASD-like Conditions in Animals

There are six different animal studies, in each of which perinatal (mostly prenatal) exposure to electronically-generated EMFs generated long-lasting neurological changes, often extending well into adulthood. Some of the effects seen clearly fall into autism spectrum-like effects. In others, there may be some doubt about that, but the pattern of EMF causation in each case is clear.

Alsaeed et al. [128] used extremely low frequency (50 Hz) EMFs to expose mice during the perinatal period, from one week before birth through to one week after birth, comparing them with controlled, unexposed mice that were otherwise treated identically. Male offspring were tested behaviorally at eight to eleven weeks of age. EMF-exposed mice showed a statistically significant lack of normal sociability, lack of preference for social novelty, and decreased exploratory activity. Other tests did not reach statistical significance. Alseed et al. [128] concluded that the EMFs studied were causally linked to ASDs. It should be noted that the [84] review showed that both extremely low frequency EMFs (including both 50 and 60 Hz), microwave frequency electronically-generated EMFs, and other diverse EMFs each acted via VGCC activation. Consequently, the Alsaeed et al. [128] findings may be relevant to microwave frequency EMF effects.

Acikgoz et al. [129] studied the effects of prenatal exposure to extremely low-frequency EMFs on three different types of ASD-like behavioral changes, and also on the levels of three proteins in the rat brain, each of which have important roles in synaptogenesis. The behavioral and biochemical changes were measured at 42 days after birth, or shortly thereafter. ASD-like behavioral changes were seen in prenatally EMF-exposed rats, but not in sham controls, using social behavior studies, open field studies, and maze tests. The three proteins studied were NLGN3, SHANK3, and SYP, each of which were found to have lowered levels in three parts of the brains, the hippocampus, amygdala, and prefrontal cortex, in EMF-exposed rats. Each of these three proteins have important roles in synaptogenesis and in synaptic function. Genetic studies have shown that each of these three proteins has a role in preventing ASDs, such that low levels of these three proteins may cause both synaptogenesis and synaptic dysfunction. Acikgoz et al. [129] also showed that the postsynaptic density was lowered by the EMF exposure. Herbert [130] has shown that lowered postsynaptic density was associated with human autism. The postsynaptic density is a macromolecular protein assembly in which various receptor pathways are physically interconnected and thereby modulate each other. These findings show that EMFs produce diverse changes in behavior and in synaptogenesis, synaptic structure, and synaptic-related protein levels, changes that are each apparently relevant to autism causation. 

Othman et al. [131,132] showed that 2 h/day of WiFi EMF exposure in utero in rats produced both neurological and oxidative stress changes in the rat brain long after birth. The WiFi radiation produced a lowered rooting reflex and righting reflex and lowered response in a rotating grid test, as well as a variable levels of acetylcholinesterase. Four markers of oxidative stress in the rat brains were each significantly elevated by prenatal WiFi exposure. 

Hong et al. [133] showed that prenatal mobile phone EMF exposure during pregnancy produced rat pups that suffered from cognitive impairment throughout their lives, into old age. The only behavioral test that was used in the Hong et al. [133] paper was the Morris water maze test, where it was shown that the elderly rats, previously exposed in utero, had distinctly lower responses than similarly aged controls that were raised under identical conditions, except their mothers had sham exposures. The hippocampi of these in utero exposed rats were studied and showed multiple major changes in their synaptic structure compared with controls. Furthermore, the expression of three genes that each control synaptic structure, SYN, PSD-95, and BDNF, were each shown to be depressed in the rats exposed in utero. When these elderly rats, exposed in utero, were moved to an enriched environment, each of these changes, including the Morris water maze test dysfunction, the histological changes in hippocampal synaptic structure, and the lowered gene expression of SYN, PSD-95, and BDNF, were each substantially, but not completely, reversed [133]. This study, which has multiple similarities to those in human ASDs, may therefore be relevant to human ASD causation. Enriched environment therapy appears to produce substantial improvements in human ASDs [134,135]. Let me state that, as a biochemist, I find it intriguing that an enriched environment produced [133] both biochemical and physiological improvements in the brain in this rat study.

Aldad et al. [136] present a paper where cell phone radiation was used to expose pregnant mice, and where the mice were examined for behavioral and neurodevelopmental changes at 8, 12, and 16 weeks after birth. The behavior was determined using videotapes of the mice, which were scored by three different observers who were each blinded with regard to the whether the mouse was or was not EMF-irradiated in utero. The mice exposed to EMF in utero were statistically significantly hyperactive, had decreased memory, and decreased anxiety, resembling human ADHD patients. The irradiated mice also had lowered miniature, excitatory postsynaptic currents, apparently due to changes in synaptic neurodevelopment. It can, of course be questioned whether the Aldad et al. [136] study on apparent ADHD is relevant to ASD causation, but it may be relevant given the many similarities between the two conditions. Kern et al. [137] suggested that ADHD may be viewed as part of the ASD spectrum. 

In the six animal studies reviewed here [128,129,131,132,133,136], each prenatal EMF exposure sequence apparently produced behavioral changes characteristic of ASDs or, in [136], ADHD. The two Othman studies [131,132] found that prenatal EMF exposure produced elevated oxidative stress long after birth, a property of human ASDs discussed elsewhere in this paper. The studies [129,133] each found lowered levels of important synapse-related proteins and changes in synaptic function, long after birth. The Hong et al. [133] study found that treating the prenatally EMF-exposed elderly rats via environmental enrichment reversed much, but not all of the behavioral and biochemical changes caused by the prenatal EMF exposures. With the exception of the last finding, each of the findings are in good agreement with predictions made in Figure 1.

### 2.6. VGCC Activation Can Act Independently of [Ca^2+^]i 

Krey et al. [138] showed that the Timothy syndrome mutation in human, rat, and mouse cells produces activity-dependent but [Ca^2+^]i-independent dendritic retraction in rodent and human neural stem cells. These experiments were performed by placing the cells in a medium containing barium and the calcium chelator EGTA to block any calcium influx through the hyperactive mutant VGCCs. Specifically, Krey et al. [138] showed that excessive free levels of the regulatory protein Gem in Timothy syndrome increases RhoA activity, which produces dendritic retraction. Are these findings of VGCC activation effects that are calcium independent surprising? Striessnig et al. [139] reviewed studies showing that both Cav1.2 (which is mutated in Timothy syndrome), and Cav1.3 VGCCs in their open channel structures, can regulate a number of specific proteins via direct protein–protein interactions, rather than acting only via increased [Ca^2+^]i. Consequently, these findings are not surprising. The Kobayashi et al. review [140] showed that depending on the conformation of the Cav1.2 channel, it can bind to Gem and therefore lower the levels of free Gem. However, as shown in [138], the Timothy syndrome mutant Cav1.2 protein fails to bind to Gem, therefore raising free Gem levels, which can produce the effects similar to those seen by Krey et al. [138]. This example shows that biology is often more complex than our attempts to model it. The findings of the [138] study suggest that VGCC activation by EMFs may be important, not just as a source of [Ca^2+^]i, but in how activation impacts synaptogenesis.

### 2.7. ASD Causation and Electronically-Generated EMF Exposures: A Summary of the Evidence

In summary, there are ten different scientists or research groups that have each proposed that EMFs may be an important cause of the autism epidemic. Each of them is influenced by the parallel course of the autism epidemic with increasing EMF exposures in diverse populations, as well as by other considerations. Parallel increases suggest, but do not prove causation.

All of the other EMF studies are based on the genetic findings, discussed in Section 1.1, showing that increased [Ca^2+^]i can cause autism/ASDs and the evidence reviewed in Section 2, showing that EMFs act primarily via increased VGCC activation and consequently increased [Ca^2+^]i. Those two findings alone make the strong prediction that EMFs can cause autism/ASDs. That prediction is greatly strengthened by the genetic studies reviewed [52,114], showing that diverse mutations that increase VGCC activity do cause autism/ASDs. Those genetic studies found cases of severe autism, such as in the case of Timothy syndrome, where there are very large increases in VGCC activity and presumably [Ca^2+^]i. They also include much lower increases in VGCC activity in the case of genetic polymorphisms which apparently cause increased autism prevalence. Such repeated genetic studies are the best approach to documenting causal roles in humans. Other important evidence on EMF causation includes repeated experimental studies on EMF impacts on neural stem cells, which inhibit neurite outgrowth from neural stem cells. Such neurite outgrowth is essential for subsequent synaptogenesis. There are five experimental studies showing that EMFs caused ASD-like effects in animal models, two of which also showed the impacts of EMF on synaptogenesis and synapse structure, and one experimental study showing that EMFs cause ADHD-like effects in animals. Epidemiological evidence rarely proves causation and the five epidemiological studies on EMFs and ASDs suggest, but do not prove, causation. But we do not need such epidemiological proof when we have such proof from human VGCC genetic studies, as well as from the experimental studies on neural stem cells and in animal models.

## 3. Organization of the Sections on the NMDA Receptors (NMDA-Rs) and Their Roles in Chemical Toxicity and Autism/ASD Causation

There are three sections on NMDA-Rs and autism causation, which are presented very briefly here; their relationship to each other and to other parts of this review can be seen in the following outline:The genetic evidence showing that NMDA-R activity is important in autism causation;The complex properties of NMDA-Rs, where an understanding of those properties is essential for understanding how toxic chemicals can act to increase NMDA-R activity and therefore cause autism/ASDs;Section 3 discusses whether and how various toxic chemicals can act to activate the NMDA-Rs or via other mechanisms to increase [Ca^2+^]i levels and therefore, potentially act to cause autism.

Please see the upper right corner of Figure 1 for relevance.

### 3.1. Mutations Changing the Activity of the NMDA-Rs Have Been Shown to Cause Autism

Ten studies have shown that mutations directly or indirectly producing changes in the activity of the NMDA receptors have roles in causing autism [141,142,143,144,145,146,147,148,149,150]. Sceniak et al. [150] and Lim et al. [144] each showed that these NMDA-R activity changes can act to produce changes in synaptogenesis. These genetic findings are similar to the VGCC genetic studies discussed above, in that both excessive activity and lowered activity can each have a role in the causation of ASD-like conditions. All of the studies on autism causation by environmental factors, to my knowledge, have shown that such factors act via increased [Ca^2+^]i. However, both VGCC and NMDA-R genetic studies show that either increased or decreased activity changes can cause autism/ASD-like effects. Deutsch et al. [147] argue for balanced allosteric control of the NMDA-Rs needed for the prevention or treatment of autism.

### 3.2. Complex Properties of NMDA-Rs

The complex properties of the NMDA-Rs have been discussed (Ransom and Deschenes [151]; Guo et al. [152]). The NMDA-Rs are glutamate receptors which differ from other glutamate receptors in that they are activated by N-methyl-D-aspartate (NMDA), rather than by other glutamate analogs which activate other classes of glutamate receptors. Although glutamate is the most important agonist of the NMDA-Rs, it is not the only such agonist. Glycine and D-serine act at another binding site on the NMDA-Rs and polyamines act at a third site. When these agonists act individually or collectively to activate NMDA-Rs, they open up a channel in the NMDA-Rs in the plasma membrane that allows calcium ions to flow into the cell and increase [Ca^2+^]i. However, that channel is blocked by a magnesium ion (Mg^2+^) under most circumstances, such that Mg^2+^ must dissociate from the NMDA-R protein before the channel can open. Such dissociation is greatly stimulated by partial depolarization of the plasma membrane, which may be produced by activation of diverse ion channels in the plasma membrane or by mitochondrial dysfunction and consequent ATP deprivation. Such dissociation is not needed if the NMDA-R channel is not blocked by Mg^2+^ because of a magnesium deficiency. Magnesium deficiency is often found in autism patients and magnesium salt supplements have been found to be helpful in autism treatment [153,154,155,156].

Homocysteine is another NMDA-R glutamate site agonist [157], and elevated homocysteine levels have been reported to be more common in autism patients than in controls [158,159,160]. Homocysteine elevation in autism patients may be due to folate or vitamin B12 deficiencies.

### 3.3. Diverse Chemicals Act Primarily but Not Solely via Increased NMDA-R Activity

The diverse chemicals thought to be important via perinatal exposure in autism/ASD causation have been reviewed [161,162,163]. These three reviews document the probable roles of volatile organic solvents, toxic metals including mercury/mercurials, lead, cadmium and arsenic, phthalates, bisphenol A, several pesticides, polyfluoroalkyl substances, and polychlorinated biphenyls (PCBs). The autism-implicated pesticides are organophosphate, carbamate, organochlorine, and pyrethroid pesticides [164].

Six of those classes of potentially autism-causing chemicals, the organophosphate, carbamate, organochlorine and pyrethroid pesticides, the volatile organic solvents, and mercury/mercurial, were reviewed regarding their mechanisms of action by the author [165] and were each found to act via NMDA-R activation (Figure 3).

Figure 3 shows that there are known pathways of action by which each of those six classes of chemicals, the four pesticides, organic solvents, and mercury/mercurials can activate the NMDA-Rs and lead to excessive [Ca^2+^]i. However, these pathways alone are insufficient to establish the importance of NMDA-R activation in producing the toxic effects of these chemicals. What was also shown in [165] was that when the toxicity of these classes of chemicals was tested in animals, the toxic effects were greatly lowered by an NMDA antagonist, showing that each of these six classes of chemicals acted largely via NMDA-R activation to produce the toxic effects. What will be shown below is that the chemicals suspected of causing ASDs each act to raise [Ca^2+^]i. While most such chemicals act by raising NMDA-R activity, three act via other mechanisms to raise [Ca^2+^]i. The mechanisms of action of other chemicals not in those six classes in Figure 3, which may also be implicated in ASD causation, are discussed below.

Another chemical apparently causing autism/ASDs is bisphenol A [161,162,163]. Bisphenol A has been shown to act via increased NMDA-R activity as shown in [166,167]. Eilam-Stock et al. [168] showed that bisphenol A reduced dendritic spine density in rats, consistent with effects that produce synaptogenesis dysfunction.

Phthalates have shown to produce effects via increased NMDA activity, and have their effects blocked by NMDA antagonists [169,170]. However, phthalates also act to inhibit the transport of the transcription factor Nrf2 into the nucleus of the cell, thus preventing its activity [171]. Because raising Nrf2 is a very important activity in treating and possibly preventing ASDs, as discussed in Section 6.1, it follows that phthalates do two things, each important in ASD causation; raising NMDA-R activity and therefore [Ca^2+^]i, and lowering the activity of Nrf2.

Valproic acid has been shown to cause hepatotoxicity, especially in very young individuals [172]. Therefore, it may act to cause autism [67,173,174,175] via hepatic encephalopathy, where liver toxicity leads to ammonia accumulation and excess ammonia acts to increase glutamatergic activity, including excessive NMDA-R activity. Ammonia does this via inhibition of the glutamate transporter, such that glutamate neurotransmission in the brain is excessive following valproic acid exposure.

Acetaminophen is thought to produce both autism and ADHD [176,177,178]. Acetaminophen overdoses can cause both hepatotoxicity and consequently, toxic encephalopathy [179,180,181]. Consequently, acetaminophen can act similarly to valproic acid, as discussed in the previous paragraph, with excessive ammonia producing excessive NMDA-R activity, leading to autism/ASDs.

Another compound that may contribute to autism causation [182] via NMDA-R activation is neomycin. Neomycin and other aminoglycoside antibiotics, such as kanamycin, are neurotoxic agents which activate NMDA-R activity by binding to its polyamine binding site [183].

Glutamate itself is an NMDA-R agonist and while glutamate is always present in the human body, ingestion of high amounts of monosodium glutamate or glutamate injection may be of concern with regard to autism causation.

The discussion shifts here to suspected ASD-causing chemicals, each acting predominantly to raise [Ca^2+^]i via non-NMDA-R dependent mechanisms. Polychlorinated biphenyls (PCBs) were listed above as one of the classes of chemicals suspected to cause ASDs. PCBs were shown in [184] to directly bind to and activate the ryanodine receptor, and therefore raise [Ca^2+^]i. The endoplasmic reticulum is the region of the cell that is most active in storing calcium ions. Activation of the ryanodine receptors on the endoplasmic reticulum membrane by PCBs releases such stored calcium ions, thus raising [Ca^2+^]i. This is, therefore, an NMDA-R independent mechanism by which PCBs can act to raise [Ca^2+^]i.

Polyfluoroalkyl substances, sometimes abbreviated as PFAS, and subgroups of these substances, designated PFOS and PFOA, are persistent contaminants in the environment, derived from some fire-fighting chemicals and non-stick cookware coatings. They have been shown to be carcinogenic, and include hepatotoxicants, neurotoxicants, and reproduction toxicants, which have been shown to cause oxidative stress. PFAS toxicity has been shown in [185] to cause “(1) changes in neurotransmitter levels, (2) dysfunction of synaptic calcium homeostasis, and (3) alteration of synaptic and neuronal protein expression and function”, each of which are consistent with ASD causation. Cao and Ng, [186] stated that “The specific mechanisms underlying such PFAS-induced neurotoxicity remain to be explored, but two major potential mechanisms based on current understanding are PFAS effects on calcium homeostasis and neurotransmitter alterations in neurons”. While specific direct effects producing PFAS neurotoxicity have yet to be determined, it is clear that elevated [Ca^2+^]i is involved in PFAS neurotoxicity, so it is plausible that PFAS may cause ASDs.

There are multiple studies that have argued for a role of aluminum ions (Al^3+^) in autism causation [178,182,187,188,189,190]. However, no case has been made for a direct mode of action by which Al^3+^-binding to a specific protein target can cause either autism or other neurotoxic responses. An exception to this pattern may be in the causation Alzheimer’s, where Al^3+^ binding to amyloid beta aggregates may have a role. Some of what is discussed here is dependent on the Skalny et al., 2021 review [189] on mechanisms of aluminum neurotoxicity, which is by far the most extensively documented review on this topic. I will only be citing three papers from that review, which are particularly important with regard to possible autism causation, but the reader may wish to look at other findings in Skalny et al. [189]. Ref. [189] describes findings that aluminum produces neurotoxicity via altered calcium homeostasis (excessive [Ca^2+^]i) and aluminum toxicity produces diverse effects including oxidative stress, neuroinflammation caused by NF-kappaB elevation and consequent elevation of inflammatory cytokines, mitochondrial dysfunction, and increased apoptosis, all of which can be produced by excessive [Ca^2+^]i (see Figure 2). The apparent centrality of excessive [Ca^2+^]i in producing aluminum-induced neurotoxicity was documented in the study by Gatta et al., [190] where a microarray of the gene expression of 35,129 genes in Al^3+^ exposed and unexposed SH-SY5Y neuroblastoma cell line cells, converted onto neuronal-like cells. The aluminum-exposed cells showed changes in gene expression almost identical to changes produced by excessive [Ca^2+^]i. Skalny et al. [189] described these findings as showing that aluminum exposure acts such that “alteration of Ca^2+^ homeostasis is the key mechanism mediating neuronal and/or synaptic dysfunction”. Aluminum action via excessive [Ca^2+^]i may also be inferred from Exley [191], who concluded that aluminum can be considered excitoxic (excitotoxicity is produced usually via excessive NMDA-R activity leading to excessive [Ca^2+^]i). These findings leave unanswered the question of how Al^3+^ exposure produces excessive [Ca^2+^]i? None of the citations in Skalny, et al. [189] answer that question. However, both Mundy et al. [192] and De Sautu et al. [193] each show that Al^3+^ binds directly to both the plasma membrane and endoplasmic reticulum (ER) calcium-dependent ATPases, inhibiting both of their activities. The plasma membrane calcium-dependent ATPase pumps calcium ions out of the cell and the ER enzyme pumps calcium into the ER, in both cases lowering [Ca^2+^]i. Interestingly, there is a very large body of literature showing that calcium-dependent ATPases in plants are inhibited by Al^3+^ and that this is the primary mechanism producing aluminum toxicity in plants. It may be concluded that Al^3+^ binding to and inhibiting calcium ATPases is the primary mechanism of aluminum cytotoxicity, and that the consequent excess of [Ca^2+^]i may, therefore, have a role in autism/ASD causation.

Glyphosate has been shown to produce increased NMDA-R activity and VGCC activity by Cavalli et al. [194] and Cattani et al. [195], with both of these papers showing that glyphosate induces a calcium influx, leading to increased [Ca^2+^]i and oxidative stress. Both VGCC calcium channel blockers and NMDA antagonists were shown to be effective in lowering effects. Gao et al. [196] showed that glyphosate acts to produce increased NMDA-R activity and excessive [Ca^2+^]i, and that NMDA antagonists can block effects caused by glyphosate. Similar mechanisms for glyphosate action were suggested earlier by Beecham and Seneff [197]. Another important study is the Cattani et al. [198] study, which used docking modeling to predict that glyphosate can bind to both glutamate and glycine/D-serine binding sites on the NMDA receptor and may, therefore, act directly as an NMDA-R agonist. However, these studies have one of the most peculiar contradictions I have ever seen in science. Cattani et al. [195] state in their abstract that “We also observed that both acute and chronic exposure to (glyphosate-containing) Roundup^®^ decreased L-glutamate uptake and metabolism …”. Taken together, these results from Cattani et al., 2014 [195] demonstrate that glyphosate-containing Roundup^®^ apparently produced lowered glutamate transport, and therefore excessive extracellular glutamate levels, and consequently, glutamate excitotoxicity, presumably via NMDA-R activation and consequent oxidative stress in the rat hippocampus. However, Cattani et al., 2017 [198] (the same research group) argue that glyphosate is acting directly on the NMDA-R protein as an NMDA agonist, and the [198] paper does so while citing the [195] paper, but fails to mention that the [198] contradicts both the interpretation and the findings of [195]. How the can we resolve this contradiction? Hultberg [199] provides evidence in favor of the [195] findings and interpretation. Hultberg [199] found that very low concentrations of glyphosate apparently greatly lowered cysteine transport, with extracellular cysteine levels being raised and intracellular cysteine and glutathione levels being lowered by very low concentrations of glyphosate. Glutathione de novo synthesis was presumably limited by low intracellular cysteine levels because intracellular cysteine is the rate-limiting precursor in glutathione de novo biosynthesis. The two most important glutamate transporters in the brain, the homologous proteins, EAAT2 and EAAT3, each transport both glutamate and cysteine. Consequently, the very low transport of both glutamate and cysteine amino acids, caused by very low concentrations of glyphosate, can be explained as being a consequence of glyphosate inhibition of one or both of these transporters. Because the charge distribution along the glyphosate molecule is similar to the charge distributions of both glutamate and cysteine, it follows that glyphosate may bind to the transport substrate binding site, acting as either a very high affinity competitive inhibitor, or as a transport blocker in EAAT2, or in both EAAT2 and EAAT3. The evidence provided here falls short of being proof of the glyphosate glutamate transport interpretation, although the evidence presented here, plus Occam’s razor, argues strongly for it. This proposed mechanism is not difficult to test. The reader may ask, what difference does this make? Either of the two mechanisms lead to the conclusion that glyphosate exposure leads to excessive NMDA-R activity. The importance of this difference is that if one argues for the wrong mechanism in the scientific, political, or legal context, we can be assured that with many billions of dollars at stake, the industry will ensure that any error will have major consequences.

In summary, 12 different chemicals or classes of chemicals act via increased NMDA-R activity to produce biological effects, apparently including ASD causation. These are four classes of pesticides, organophosphates, carbamates, organochlorine and pyrethroid pesticides, volatile organic solvents, mercury/mercurials, bisphenol A, phthalates, acetaminophen, neomycin/aminoglycoside antibiotics, glutamate, and glyphosate. All 12 of these produce excessive [Ca^2+^]i, as do three other classes of chemicals also thought to be possible causes of ASDs: polyfluoroalkyl substances (PFAS), where the mechanism of action is unknown; aluminum ions (Al^3+^), which bind to and inhibit the activity of calcium ATPases, and polychlorobiphenyls (PCBs), which bind to the ryanodine receptor and cause calcium ions to leak out of the endoplasmic reticulum into the main part of the intracellular space.

I do not think that the fact that each of these 15 chemicals/classes of chemicals each act to produce excessive [Ca^2+^]i is coincidental. The central focus of both ASD causation and the autism/ASD epidemic is [Ca^2+^]i. These findings are, therefore, important evidence that the autism/ASD epidemic is real, and that these chemicals have important roles in causing the epidemic. It is essential to look at the strongest available evidence, rather than just epidemiology, when assessing the reality of the autism epidemic.

Additional points: Phthalates act in a second way to cause ASDs. They inhibit the transport of the transcription factor Nrf2 into the nucleus, thus blocking Nrf2-dependent therapeutic effects in ASDs (see Section 6.1). There have been two proposed pathways of action for glyphosate to produce increased NMDA-R activity, with this author arguing for one of those pathways and against the other. The genetic studies that have shown that increased NMDA-R activity causes ASDs further strengthens the causal argument for each of the 12 classes of chemicals, which each act via increased NMDA-R activity.

Chemicals acting via excessive NMDA-R activity should be viewed as an important new paradigm in chemical toxicology. Having 12 classes of chemicals raising NMDA-R, and consequently producing excessive [Ca^2+^]i via somewhat different mechanisms, raises the possibility of possible synergistic toxicity among them. The actions of aluminum hydroxide/Al^3+^ and PCBs, which each act to raise [Ca^2+^]i via different mechanisms from those 12 classes of chemicals, predict synergistic toxicity between each of these classes of chemicals, and also between each of the two and those 12 classes of chemicals that act via NMDA-R activation. Therefore, these considerations may make predicting the neurotoxicity of combinations of these chemicals especially challenging. We also have, in addition, possible chemical synergism with EMFs acting via VGCC activation. We live in a neurotoxic soup, the effects of which are currently impossible to predict based on the individual toxicities of the individual components.

### 3.4. Agents That Lower VGCC Activity or NMDA-R Activity Are Reported to Produce Protective Effects in Autism Rat Models

Kumar et al. [174] found that the VGCC calcium channel blocker nimodipine was useful in lowering both biochemical and behavioral effects in a valproic acid-induced autism rat model. The protocol they used was to induce autism by using a single injection of valproic acid at day 12.5 after conception and at 24 days after birth, to use I.P injection of nimodipine once a day. The valproic acid-induced biochemical changes that were greatly normalized by nimodipine included two markers of oxidative stress, three inflammatory cytokines, and changes in both CREB and BDNF activity [174]. The behavioral changes were also lowered by nimodipine, including three measures of social interactions [174].

Demirkaya et al. [200] reported that two drugs, riluzole and gabapentin, were useful in improving spatial learning, locomotor activity, anxiety, and social behaviors in a rat autism model. Riluzole lowers glutamate release in the brain, therefore lowering NMDA-R activity, whereas gabapentin lowers VGCC activity. Consequently, there is evidence from studies in a rat autism model that the two main environmental pathways outlined in Figure 1, which produce increases in [Ca^2+^]i, may each be implicated in producing the widespread effects of ASDs, not in the initial causation, but rather in the chronic subsequent condition.

## 4. GABA_A_ Switching during the Early Post-Natal Period—Another Issue That Makes ASD Causation More Complex

There is one complication with the GABA_A_ pathway (Figure 3) as it applies to the prenatal and very early postnatal period. As stated in Cherubini et al. [201] review “In the prenatal and immediate postnatal period GABA, acting on GABA_A_ receptors, depolarizes and excites targeted cells via an outwardly directed flux of chloride. In this way it activates NMDA receptors and voltage-dependent calcium channels contributing, through intracellular calcium rise, to shape neuronal activity and to establish, through the formation of new synapses and elimination of others, adult neuronal circuits”. GABA_A_ receptors act in the prenatal/very early postnatal period oppositely from how they work later. They further state that “Thus, the premature hyperpolarizing action of GABA or its persistent depolarizing effect beyond the postnatal period, leads to behavioral deficits associated with morphological alterations and an excitatory (E)/inhibitory (I) imbalance in selective brain areas”. Because GABA_A_ receptor activation opens a chloride channel, the switch here in GABA_A_ receptor action is dependent upon a switch in the way intracellular chloride concentrations are maintained, which determines, in turn, whether GABA_A_ receptor activation is depolarizing (early perinatal) or hyperpolarizing (later development into adulthood). Cherubini et al. [201] spend circa two pages in their review documenting the roles of both premature switching and delayed switching in certain regions of the brain, with both causing both ASDs. Depolarization produces increased [Ca^2+^]i largely via increased VGCC activity, whereas hyperpolarization produces decreased [Ca^2+^]i, largely via decreased VGCC and NMDA-R activity. Consequently, this GABA_A_ switching mechanism suggests that somewhat higher [Ca^2+^]i is required during the late prenatal and very early postnatal period in order to achieve normal synaptogenesis. Neuronal migration, which requires elevated [Ca^2+^]i, is especially active during that period of development (Hwang et al. [202]; Ang et al. [203]). The requirement for [Ca^2+^]i elevation for neuronal migration may therefore provide a partial explanation for how delayed switching of GABA_A_ causes ASDs. This may also explain how some genetic studies, showed that lowered VGCC and NMDA-R activities each produced increased ASD causation. Both low [Ca^2+^]i as well as high [Ca^2+^]i can each cause ASDs (Section 2.2 and Section 3.1).

The findings in the Ang et al. [203] study showing that ultrasound exposure can disrupt neuronal migration suggest a possible role of ultrasound in ASD causation, a suggestion that has been controversial in the ASD literature.

## 5. Mutations Having Major Roles in Causing Forms of Autism: Focus on the Mechanism of Autism Causation

Genetics are a powerful methodology for determining biological mechanisms; when mutations in specific genes repeatedly produce a biological response, such findings are very powerful in proving that biological mechanisms involving those genes produce that response. Section 2.2, Section 3.1, and Section 4 of this paper have each shown that mutations which produce changes in [Ca^2+^]i levels do cause ASDs.

The goal in this section is to look at reviews of mutations that each have major roles in causing autism/ASDs, in order to determine, in a broader sense, what those mutations tell us about autism/ASD causation. Syndromic forms of autism are caused by specific gene mutations that produce autism which has behavioral and or physiological properties that differ from those commonly found in the broader autism population. By looking at mutations producing both syndromic and non-syndromic forms of autism, we can get important insights into autism/ASD causation. These mutations in the genes include both point mutations and copy number mutations. Such mutations have major documented causal roles in circa 12 to 25% of autism cases.

The Huguet et al. [204] review summarizes their findings as follows: “In summary, the genes associated with ASDs are numerous and involved in multiple cellular functions including chromatin remodeling, metabolism, mRNA translation and synaptic function. The downstream consequences of the mutations, however, might converge to a defect in neuronal/synaptic homeostasis”.

In the Crawley [205] review, mutations producing ASD-like effects fall into three categories, as shown in Table 1 in [205]: synaptic cell adhesion genes with roles in both synaptogenesis and synaptic function, signaling and developmental proteins, and neurotransmitters and their receptors. 

Persico and Napolioni [206] divide such mutations into five categories. In the first of them, on synaptic genes, Persico and Napolioni, 2013 state: “Several neuroligins, SHANK and neuroxin genes, *encoding proteins crucial to synapse formation maturation and stabilization*, have been found to host mutations responsible for behavioral phenotypes, including autism (italics added). At the extracellular level, postsynaptic neuroligins interact with presynaptic a- or b-neurexins stimulating the formation of the presynaptic bouton; at the intracellular level, neuroligins associate with post-synaptic scaffolding proteins, such as SHANK” (citations omitted and italics added). These findings are of great importance because they specifically implicate aberrant synaptogenesis in autism causation.

The second category is chromatin architecture genes. Here, they concentrate on one example, MECP2. Persico and Antonioni [206] state: “The MECP2 gene is, thus, required for correct brain function and development; loss of MECP2 has been shown to delay neuronal maturation and synaptogenesis”. Because the MECP2 enzyme is inhibited by [Ca^2+^]_i_, excessive [Ca^2+^]_i_ may be predicted to produce effects similar to mutational loss of MECP2. While these findings fall short of proof that MECP2 mutations are acting mainly via their impacts on synaptogenesis, they are consistent with that mechanism of action.

The third such category is morphogenetic and growth-regulating genes, which contain three genes, HOXA1, PTEN, and EIF4E. These each produce multiple effects on development, but each include brain development in their spectrum of activities. PTEN knockout mice display “macrosomy macrocephaly CNS overgrowth with thickening of the neocortex and cytoarchitectonic abnormalities in the hippocampus, *excessive dendritic and axonal growth and increased numbers of synapses*” (italics added). EIF4E has both gain of function and loss of function mutations implicated in autism. Transgenic mice overproducing the gene product eIF4E display “enhanced excitation/inhibition ratio, increased translation of neuroligins, and autistic behavior”. This third mutation category [206] is entirely consistent with the model presented in this paper, but because of the diversity of effects produced by these gene mutations, it does not provide compelling evidence for that model.

The fourth category in Persico and Antonioni [206] is calcium-related genes, focusing on the VGCC genes as outlined elsewhere in this paper, and also genes that indirectly lead to increased calcium influx. These findings directly implicate increased [Ca^2+^]i in autism/ASDs and therefore provide strong support for the part of the model presented Figure 1.

With regard to the fifth (mitochondrial) category, the authors state that “Biochemical parameters linked to mitochondrial function are frequently abnormal in autism. However, mitochondrial dysfunction appears secondary to the pathophysiology underlying ASDs in the vast majority of cases”. They do go on to describe very rare mitochondrial mutations causing ASDs, which act to produce lowered ATP synthesis and/or increased mitochondrial superoxide production. These findings may be interpreted in terms of the mitochondrial dysfunction role of ASDs and the probable role of the NO/ONOO(-) cycle (see Section 6.1).

In summary, the second, third, and fifth categories of autism/ASD mutations described in Persico and Antonioni [206] are consistent with the autism model presented in Figure 1, but none of those three provide any compelling evidence for that model because of the breadth of the effects of mutations produced by those types of mutations. In contrast, category 1 mutations that act directly to disrupt synaptogenesis, and category 4 mutations that directly produce very large changes in calcium influx and therefore large changes in [Ca^2+^]i, together provide compelling evidence for the central focus of the Figure 1 model. Specifically, together they provide compelling evidence for the proposed mechanism, where excessive [Ca^2+^]i acts to disrupt synaptogenesis during the perinatal period. The Huguet and the Crawley reviews [204,205], with their mutational focus on synapses including synaptogenesis, also support the central part of the Figure 1 model.

## 6. The NO/ONOO(-) Cycle and the Chronic Nature of Autism/ASDs

The author first proposed a simplified version of the NO/ONOO(-) cycle as the central cause of chronic fatigue syndrome [207], and then later as a more complex cycle, as the central cause of many diseases [208]. The most extensive description of the cycle was made in a 57 page paper on heart failure, where the most detailed description of the NO/ONOO(-) cycle was made, and the cycle was shown to be the central cause of heart failure [209]. A similar vicious cycle mechanism has been proposed by Dr. Guy C. Brown and his colleagues as the central cause of neurodegeneration [210,211]. The elements of the NO/ONOO(-) cycle all act at the level of individual cells, such that the cycle can be, at least in principle, the cause of large numbers of diseases, depending on which tissues of the body are most impacted [209]. In the case of autism/ASDs, the time in the lifecycle during which the NO/ONOO(-) cycle may start may be of critical importance in distinguishing autism/ASDs from other NO/ONOO(-) diseases primarily impacting the brain. All vicious cycles have the property that each element in the cycle is both a cause and an effect of the cycle, so linear logic cannot be used to understand them. At this point in the paper, no evidence has been given that the NO/ONOO(-) cycle has a role in autism/ASDs, but it is necessary to examine the cycle before any such evidence can be interpreted by the reader. It is proposed that both of the pathophysiological pathways of action in Figure 2 have roles in NO/ONOO(-) cycle initiation, but that the peroxynitrite/free radical/oxidative stress/NF-kappaB/inflammatory cytokine pathway is especially important.

The NO/ONOO(-) cycle is outlined in Figure 4. The cycle is made up of five individual vicious cycles which are, in turn, based on 34 different mechanisms each acting at the level of individual cells [209]. Peroxynitrite (ONOO(-)) is the only element in the cycle that is an element in each of those five, such that it can be considered most central to the entire NO/ONOO(-) cycle. There are however, four other centers, each with key importance for the entire cycle. On the right side of Figure 4 is the inflammatory part of the cycle, focused on NF-kappaB elevation and elevation of the inflammatory cytokines. The red arrows leading into and out of those inflammatory regulators indicate some of the key regulatory mechanisms. There is a double-headed arrow linking the five cytokines to NF-kappaB because three of those five cytokines act to raise NF-kappaB.

A large number of studies have shown that NO/ONOO(-) elements are each raised in autism/ASDs (Table 1) [206,207,208,209,210,211,212,213,214,215,216,217,218,219,220,221,222,223,224,225,226,227,228,229,230,231,232,233,234,235,236,237,238,239,240,241]. 

**Table 1 brainsci-14-00454-t001:** NO/ONOO(-) elements that are raised in autism/ASDs.

Peroxynitrite (ONOO-)	Peroxynitrite was shown to be elevated in autism based on a test using 3-nitrotyrosine as a specific marker for peroxynitrite (Sajdel-Sulkowska et al. [213]; De Felice et al. [214]; Nadeem et al. [215,216]; Algahtani et al. [217]; Carey et al. [218]). Peroxynitrite elevation by the NO/ONOO(-) cycle, as discussed here and in thousands of other papers, produces oxidative stress. Peroxynitrite is scavenged through its reaction oxidizing 5-methyltetrahydrofolate (5-MTHF) [219,220], thus depleting cells of a very important methyl donor, while lowering peroxynitrite. These properties of peroxynitrite mean that both the oxidative stress and the low methylation status of autism patients reported by James et al. [221] may be a consequence of peroxynitrite elevation.
Oxidative stress	See above. There are dozens of papers showing oxidative stress in people with autism/ASDs, as well as in animal models. The following examples each provide strong evidence for an oxidative stress role in ASD causation: Carey et al. [218]; De Felice et al. [214]; Golomb [212]; James et al. [221]; McGinnis [94]; Nadeem et al. [215,216]; Saidel-Sulkowska et al. [213], and Bjorklund et al. [222].
NF-kappaB	NF-kappaB has been among the less studied of the measurable elements of the cycle in autism. NF-kappaB has been found to be elevated in cases of autism in Young et al. [223]; Nadeen et al. [216]; Liao and Li [52], and Theoharides et al. [224]. Young et al. [223] may be of special interest, in that the paper was completely focused on NF-kappaB measurements. They used an immunoassay for NF-kappa that can be scored under a microscope, studying post-mortem orbitofrontal cortex brain tissues of patients with ASDs and matched normal controls. Young et al. [223] found very highly statistically significant elevation of NF-kappaB in ASD vs. controls in each of the three cell types studied, neurons, astrocytes, and microglia. They also found that there were similar elevations in both the nuclear localized and extranuclear NF-kappaB in ASD tissues. Because the NO/ONOO(-) cycle predicts cycle elevation of the synthesis of NF-kappaB, these findings are in good agreement with prediction. Alomar et al. [225] showed that NF-kappaB elevation has a causal role in a mouse model of autism.
Inflammatory cytokines	Eftekharian et al. [226] found that each of the five inflammatory cytokines predicted to be directly raised by NF-kappaB elevation were raised in autism patients, as well as several other cytokines. Theoharides et al. [224], Kutuk et al. [227], and Ferencova et al. [228] found, similarly, in their large studies, that the same five inflammatory cytokines directly raised by NF-kappaB were elevated in patients with ASDs, as were multiple additional cytokines. Mehta et al. [229] showed that four of those five cytokines (all except IFNγ) were elevated in a rat model of ASD.
Mitochondrial dysfunction	Mitochondrial dysfunction has been demonstrated in individuals with autism/ASDs. Napolioni et al. [230] ascribed much of the mitochondrial dysfunction seen in those with ASDs to the dysfunction of the calcium-regulated aspartate/glutamate carrier. Others reviewing mitochondrial dysfunction in autism include Golomb [212], Rossignol and Frye [231], and Palmieri and Persico [232]. When Palmieri and Persico [232] asked whether mitochondrial dysfunction is either the cause or effect of ASDs, they concluded it is both, consistent with the NO/ONOO(-) cycle predictions that all cycle elements must be both cause and effect.
[Ca^2+^]i	Autism/ASDs caused by excessive [Ca^2+^]i has been documented throughout this paper.
Tetrahydrobiopterin (BH4) depletion	Tani et al. [233] and Frye [234,235] have each demonstrated a deficiency of tetrahydrobiopterin (BH4) in autism patients. A series of studies have each shown that BH4 supplementation is useful in autism treatment (Naruse et al. [236]; Takesada et al. [237]; Frye [234]). The NO/ONOO(-) cycle produces, as part of the cycle, peroxynitrite-mediated oxidation of BH4 to dihydrobiopterin (BH2), thus lowering BH4 levels. Consequently, BH4 supplementation will not only provide the BH4 cofactor for several enzymes, including the nitric oxide synthases, lowering the NO/ONOO(-) cycle in that way, but will also lower peroxynitrite by chemically reacting with it, lowering the cycle in a second way.
iNOS	iNOS has been found to be elevated in patients with ASDs (Nadeem et al. [215,216]). Mehta et al. [226] showed that iNOS was elevated in a rat model of ASDs.
Nitric oxide (NO)	Several reviews documented elevated nitric oxide in patients with ASDs, including Nadeem et al. [215,216] and Tripathy et al. [238]). Mehta et al. [229] showed that nitric oxide was elevated in a rat model of ASDs.
RhoA	In the paper showing that the NO/ONOO(-) cycle is central to the etiology of heart failure, it was shown that RhoA functioned as possibly tissue-limited cycle element [209]. RhoA is implicated as a causal element of autism/ASDs, including in stabilization of synapses (Richter et al. [239]; Luo et al. [240]. Hayashi et al. [241]), providing further evidence consistent with a NO/ONOO(-) cycle role in autism/ASDs.

Another important center is [Ca^2+^]i, which is closely linked to the calcium ATPases. There are two distinct calcium ATPases, one of which is in the plasma membrane, which uses the energy of ATP hydrolysis to pump calcium ions out of the cell, thus lowering [Ca^2+^]i. The other is located endoplasmic reticulum membrane and pumps calcium from the intracellular space into the endoplasmic reticulum. Both of the calcium ATPases are inhibited by mitochondrial dysfunction due to their requirements for ATP, and both are inactivated by peroxynitrite. The endoplasmic reticulum calcium ATPase is thought to be most important in controlling [Ca^2+^]i over very short times, whereas the plasma membrane enzyme is much more important in controlling [Ca^2+^]i over longer times. In Figure 1, [Ca^2+^]i is linked to the NO/ONOO(-) cycle mechanism via a double headed arrow. [Ca^2+^]i as a cycle element raises the cycle and the cycle, in turn, raises all cycle elements, including [Ca^2+^]i. There are also three classes of plasma membrane calcium channels, each proposed to have increasing activity as part of the cycle: the VGCCs, the NMDA-Rs, and multiple types of TRP receptors including, but not limited to, the TRPA1 and TRPV1 receptors.

Mitochondrial dysfunction is an additional type of dysfunction, leading to both lowered ATP synthesis and increased superoxide production from the electron transport chain.

The last of the five regulatory centers of the cycle is shown in Figure 4 by green arrows going in both directions between ONOO(-) and tetrahydrobiopterin (BH4) depletion. The peroxynitrite, ONOO(-) oxidizes BH4 to BH2, which is often further oxidized and metabolized into compounds that are no longer in the biopterin pool and leads, therefore, to longer-term BH4 depletion. BH4 is a cofactor on the nitric oxide synthases (NOSs), such that when the NOSs have no BH4 bound to them, they are said to be uncoupled and the uncoupled NOSs synthesize superoxide (OO·) in place of nitric oxide (·NO). Superoxide and nitric oxide are the two precursors of peroxynitrite. Consequently, when there are high concentrations of NOSs in a region of the cell and some are coupled (BH4 bound), and others are uncoupled because of peroxynitrite oxidation of BH4, many nearby NOS enzymes can simultaneously be generating the two precursors of peroxynitrite, making the synthesis of peroxynitrite particularly high.

Ten measurable elements of the NO/ONOO(-) cycle have been measured in human patients with ASDs and/or in animal models of ASD, and each of the ten have been shown to be elevated relative to controls (Table 1) [212,213,214,215,216,217,218,219,220,221,222,223,224,225,226,227,228,229,230,231,232,233,234,235,236,237,238,239,240,241]. It follows from this that the cycle is elevated and may, therefore have a causal role in autism/ASDs, such that it may be predicted to produce, in part, the chronic nature of ASDs. That causal role is further confirmed in the next section of this paper.

### 6.1. The NO/ONOO(-) Cycle and the Chronic Nature of Autism/ASDs: Therapeutic Approaches by Raising Nrf2 and AMPK

The NO/ONOO(-) cycle mechanisms, as shown in Figure 4, are what happens when all of the many positive feedback loops take over in a particular tissue, or tissue regions, despite whatever homeostatic mechanisms may be present to oppose them. Two such homeostatic mechanisms are discussed in this section, where the levels of each of them can be raised by health-promoting factors, which may act, in turn, to lower the NO/ONOO(-) cycle and thus produce improvements in cycle-caused diseases. The question being raised here is whether raising either or both Nrf2 or AMPK is useful in the treatment of autism/ASDs.

Let us consider Nrf2 first. Nrf2 is a transcription factor that controls the transcription of large numbers of genes encoding cytoprotective enzymes. It is best known for its role in raising levels of many antioxidant enzymes [242,243,244]. It raises levels of superoxide dismutases and catalase, and raises the levels of reduced glutathione and glutathione peroxidases. Nrf2-raised glutathione peroxidase and Nrf2-raised peroxyredoxins each bind to and lower peroxynitrite levels. Raising Nrf2 also lowers the levels of the two precursors of peroxynitrite, superoxide and nitric oxide. Because, as discussed in the previous section, peroxynitrite is central to the NO/ONOO(-) cycle mechanism, lowering peroxynitrite alone may be sufficient to down-regulate the NO/ONOO(-) cycle. Raising Nrf2 also lowers NF-kappaB activity and consequent inflammation and improves mitochondrial function, while increasing detoxification mechanisms which detoxify the body of both carbon-containing toxicants and toxic metals [244,245,246]. Each of these actions may be predicted to lower the NO/ONOO(-) cycle and predict consequent improvements in chronic diseases, where the cycle may act to maintain their chronicity. These consequences of raising Nrf2 are what lead to the inference that [244] that “Raising Nrf2 may be nature’s way of preventing or treating NO/ONOO(-) cycle diseases”, as follows: Figure 5, taken from [244] with permission, outlines each of these mechanisms with regard to raising Nrf2.

Yang et al. [245] conducted a systematic review of the literature on using agents raising Nrf2 to treat ASDs. Their research group is well informed about the roles of Nrf2, not only in lowering oxidative stress but also in lowering mitochondrial dysfunction and NF-kappaB and inflammation. They state, in their abstract “Ultimately, we identified 22 preclinical studies, one cell culture study, and seven clinical studies, covering a total of five Nrf2 activators. For each Nrf2 activator, we focused on its definition, potential therapeutic mechanisms, latest research progress, research limitations, and future development directions. Our systematic review provided suggestive evidence that Nrf2 activators have a potentially beneficial role in improving autism-like behaviors and abnormal molecular alterations through oxidant stress, inflammation, and mitochondrial dysfunction”.

Most of my discussion here on specific studies is focused on six more recent studies, that were not reviewed by Yang et al. [245]. However, three articles reviewed in [245] of apparent special interest are discussed here, each of which suggest that raising Nrf2 during gestation in a rat autism model may ameliorate or even prevent autism/ASDs.

The first of these is the Fontes-Dutra et al. [247], a study which looked at whether raising Nrf2 can help ameliorate the severity or occurrence of ASD via treatment of pregnant female Wistar rats, where ASD was caused by valproic acid injection of pregnant females at day 12.5 after conception. To raise Nrf2, the authors used resveratrol injected once a day each day, from day 6.5 to 18.5 after conception. While the authors found that there were changes, apparently relevant to the ASD, that were largely normalized by resveratrol injection, they also found other such changes where resveratrol did not produce a statistically significant improvement. Fontes-Dutra et al. [247] measured three behavioral effects related to nest-seeking behavior, one of which was greatly improved by resveratrol, but the other two were not. Two tests were performed on whisker nuisance task and both of those showed statistically significant improvements from resveratrol injection. The other tests that were done were on valproic acid-induced histological structural changes in regions of the brain, where resveratrol did ameliorate those changes. They found that valproic acid injection produced a deficiency in GABAergic neurons in the primary sensory area in the brain, which was largely prevented by the resveratrol injection. This valproic acid-induced effect was attributed by the authors as being due to defective neuronal migration. They also found that the levels of the synapse-related protein gefrin was depressed by valproic acid treatment in the primary sensory region of the brain, and that the depressed gefrin levels were largely blocked by resveratrol. While it is clear from [247] is that resveratrol did not prevent all valproic acid-induced effects, it did produce, in the pregnant mice, very substantial improvements in diverse effects that may be involved in causing ASDs in the offspring. While the authors expressed the view that resveratrol may not be “the ideal agent” to raise Nrf2, this is an important study that suggests that mothers may be able to lower the occurrence or the severity of ASDs in their babies by using agents, including nutritional agents, that raise Nrf2.

The Hirsch et al. [248] study was performed using the same methodology as in [247] with valproic acid and resveratrol, except that different ASD-related end points were scored. They showed that injecting resveratrol during pregnancy improved four valproic acid depressed social interaction end points in the offspring. They also found that valproic acid-induced rises in the micro-RNA miR134-5p in the offspring were also blocked by in utero resveratrol. However, measures of self-grooming behavior impacted by valproic acid were not statistically significantly affected by resveratrol. Consequently, as in [247], ref. [248] sees amelioration of several, but perhaps not all, effects.

The Dai et al. [249] article used a similar methodology to that used in [247,248]. The authors found that valproic acid produced “excessive repetitive behavior, lower social interaction, longer moving time and distance in central area, and reduced standing times”. However, “resveratrol treatment corrected the VPA-induced autistic-like behaviors (*p* < 0.05)”. The findings in [247,248,249] show that the Nrf2 raising agent resveratrol, when given during pregnancy, ameliorates diverse but not all effects produced by the ASD-causing agent valproic acid. Furthermore, the experimentally found amelioration is consistent with the known properties of Nrf2 and the roles of the NO/ONOO(-) cycle in ASD causation, as shown in Table 1. These findings suggest that well-tolerated agents that raise Nrf2 should be tested in humans for their possible role as agents that lower the severity, and possibly the occurrence, of autism/ASDs.

Six recent studies, not covered by the Yang et al. [245] review, are discussed below.

The study by Subasi Turgat et al. [250] is a human case control study where the levels of the Nrf2 protein and its binding protein, Keap1, were measured in ASD patients and normal control patients. When Nrf2 is bound to Keap1, the Nrf2 protein undergoes rapid proteolysis, such that the levels of both of these two proteins are important in regulating Nrf2 function. Subasi Turgat et al. [250] found that ASD patients had lower levels of Nrf2 and higher levels of Keap1 than normal controls, suggesting but not proving that lowered Nrf2 may have a role in ASD causation. Such causation is also suggested by the Porokhovnik et al. [251] study, in which there were no significant differences shown in autism incidence in patients carrying a genetic polymorphism in the Nrf2 gene compared to those carrying the more common Nrf2 gene, but some significant differences in oxidative DNA effects and another stress response in ASD patients were shown. That study, although an interesting one, may be limited because of its insufficient size.

Wu et al. [252] used selenium in the form of sodium selenite to raise Nrf2 in a BTBR mouse model of ASD. They showed that selenite greatly lowered both ferroptosis in the mouse hippocampus, and also “significantly mitigated impairments in learning and memory, improved social functions, reduced repetitive behaviors”. These effects of selenite were blocked by using a Nrf2 inhibitor, showing that the Nrf2 pathway was essential for producing the selenite-induced effects. This study therefore provides strong evidence that raising Nrf2 lowers ASD effects in an ASD mouse model. This study and the others discussed below are treatment studies, not prevention or in utero amelioration studies such as [247,248,249].

Shah et al. [253] present a review that explores the possibility of using sulforaphane to raise Nrf2 and down-regulate the oxidative stress, inflammatory and mitochondrial dysfunction found in ASDs. The Shah et al. [253] review takes a position on these topics that is very similar to the position taken here, with the exception being that they only looked at sulforaphane, which is only one of many nutritional and non-nutritional agents that raise Nrf2.

Abbasalipour et al. [175] used sumac and gallic acid-loaded nanophytosomes to raise Nrf2 and ameliorate hippocampal oxidative stress in a valproic acid-induced rat model of ASD. Six different biochemical changes that are characteristically raised in the hippocampus of ASD patients, were each lowered by raising Nrf2 in the rat hippocampus in this rat ASD model [175]. The only cognitive change analyzed in [175] was recognition memory, which also showed substantial improvement from this Nrf2-raising treatment.

Zhao et al. [254] studied the effects of raising Nrf2 in a valproic acid-induced rat ASD model. They used electroacupuncture on the heads of the pups to raise Nrf2 and showed that electroacupuncture “ameliorated the locomotor activity, social behavior, spatial learning and memory and repetitive behavior compared with control ASD rats”, while also producing several biochemical changes known to be produced by raising Nrf2. However, the effects induced by electroacupuncture were each abolished after injection of an adenoviral vector directing the production of small interfering RNA, which interferes with Nrf2 biosynthesis. The small interfering RNA technology is known to be very highly specific. Consequently, the improvements in “the locomotor activity, social behavior, spatial learning and memory and repetitive behavior” produced by electroacupuncture were produced by raising Nrf2. If you search the PubMed database for electroacupuncture and Nrf2, you will find 34 papers where electroacupuncture raises Nrf2, one of which is [254]. These findings [254] are, therefore, the most convincing studies showing that raising Nrf2 clearly produces therapeutic improvement in an ASD animal model. Therefore, using agents to raise Nrf2 should be viewed as a promising approach to treating human ASD patients.

AMPK is another regulatory system which lowers elements of the NO/ONOO(-) cycle and, therefore, possible cycle involvement in autism, which may predict that agents that raise AMPK may be helpful in the treatment of autism/ASDs. AMPK is a protein kinase which is best known for improving mitochondrial function. However, it also has established roles in lowering NF-kappaB and lowering inflammation, and is reported to lower superoxide production and therefore, oxidative stress in mitochondria. There are several studies suggesting that agents raising AMPK may be useful in autism/ASD treatment (Bakoui [255]; Alsaqati et al. [256]; Lanfranchi et al. [257]; Diaz-Gerevini et al. [258]; Gantois et al. [259]).

## 7. Discussion and Conclusions

Various parts of Figure 1 that have each been documented in this review as important causal elements in autism/ASDs are not new.

The special role of perinatal events is not new;The essential role of elevated [Ca^2+^]i and consequent calcium signaling is not new;The special roles of synaptic changes and synaptogenesis is not new;Even the roles of depressed levels of [Ca^2+^]i is not new;The importance of EMF exposures is not new, although the detailing of diverse types of supportive evidence is new;The importance of chemical exposure and even the action of diverse chemicals via increased [Ca^2+^]i is not new;The NO/ONOO(-) cycle as an apparent cause of chronic disease is not new, although documentation of its role in autism/ASDs is, to my knowledge, new;The role of raising Nrf2 in the prevention or treatment of apparent NO/ONOO(-) cycle diseases is not new, nor is it possible role in treatment or prevention of ASDs;Mutational causation of autism/ASDs via mutations causing changes in [Ca^2+^]i or in synaptogenesis is not new.

Despite all this pre-existing science, Figure 1 does provide a novel understanding of autism/ASD causation, a novel understanding that is extensively documented in this review.

Figure 1, however, does not fully describe the causal mechanisms involved in ASD causation:It does not include number 4 above, that abnormally low [Ca^2+^]i also can cause autism/ASDs;It does not include the finding in [138] that VGCCs can have roles in ASD causation independent of [Ca^2+^]i, although, as discussed in the text, this can be produced via direct protein–protein interactions;It was correctly pointed out to the author by one reviewer that glial cells [260,261,262,263], including astrocytes [264,265,266,267,268], have roles in synaptogenesis, but only the roles of neurons were included in Figure 1. Furthermore, glia, including astrocytes, have NMDA-Rs [269,270], as well as the almost universally found VGCCs, and glial calcium signaling is of great importance.

It is often the case that the biology is more complex than our models—these are each examples of that.

Let us shift now to ASD treatment and prevention. These are presented here not to give medical advice, but rather to enlighten the reader in the central issues of this special issue: *The current situation and prospects for children with ASDs.* Both treatment and prevention require avoidance of electronically-generated EMFs and diverse chemicals, however difficult that may be. Let us start with EMFs. There are many sources of EMFs in our living spaces, workplaces, educational and recreational environments, and even outdoors. Many of these are discussed in Section 2.1.

There are “safety guidelines” initially proposed by ICNIRP, an organization set up by the industry and these ICNIRP safety guidelines have been adopted, sometimes with minor modifications, over much of the world. The ICNIRP and similar “safety guidelines” are all based on thermal effects produced over either a 6 min or 30 min period. However, as discussed above, biological effects are caused by electric forces or time-varying magnetic forces produced by electronically-generated EMFs, and are unrelated to thermal effect “safety guidelines”. It follows that all assurances of safety based on these “safety guidelines” have no scientific merit.

Pregnant women should not use cell phones, cordless phones, should avoid Wi-Fi fields, and should use wired connections to the internet, avoid Bluetooth, should not spend time near smart meters, cell phone towers, avoid high voltage powerlines, etc. Pregnant women should avoid being in buildings with high dirty electricity in their electric powerlines. Such avoidance is very challenging. A second choice is to use shielding to protect the fetus and the young baby after birth. 

You can get large shielded T-shirts to protect the fetus, wearing a regular cloth T-shirt underneath to minimize body contact. Such shielding materials have large numbers of metal (often silver) fibers. I believe that shielding mostly acts by disrupting the coherence of the EMFs, therefore greatly lowering any electric or magnetic forces. Shielding has been extensively studied in the scientific literature [271,272,273,274,275,276,277]. For example, ref. [271] states that “By using wireless technologies, we increase the influence of the electromagnetic field on the human body. The basic measures for the protection against electromagnetic radiation are: (1) shielding of the radiation source, (2) shielding of the workplace, (3) personal protective equipment, (4) reduction of radiation in the immediate vicinity of the radiation source”. [271] goes on to discuss the shielding effects of both fabrics containing tiny metal fibers and also brick walls.

You can buy shielding paint of two different types to protect yourself in your house and, if you live in an apartment, this may be the only way to protect yourself from your neighbors’ EMFs. Shielding paint can be either graphite (black) paint which can be painted over with other colors. Other more pleasant colored shielding paint can be purchased containing tiny metallic particles. You can also buy open shielding cloth to put over windows, making them into lace like curtains to lower EMF exposures coming through the windows. Similar approaches can be used after birth to protect the young baby. It is highly desirable to purchase a good EMF meter to provide information and prevent the worst exposures throughout the perinatal period.

The materials mentioned here, along with many others, can be obtained from LessEMF, an organization with which I have no commercial relationship—I have no conflict of interest here.

Magnesium supplements, vitamin D, and folates are reported to be helpful for both prevention and treatment. Raising Nrf2 was discussed as a very promising approach to ASD therapy and possibly prevention, because raising the levels of the Nrf2 transcription factor produces increases in very wide-ranging antioxidant enzymes, and also lowers NF-kappaB and inflammation, improves mitochondrial function, and also detoxifies the body of diverse carbon-containing toxicants and toxic metals. There is much evidence that raising Nrf2 is useful in ASD treatment in animal models, as well as in prevention or amelioration, when Nrf2 is raised in pregnant animal models. The use of therapeutic agents that raise Nrf2 during pregnancy must, of course, be used with great care. There is a large body of safety literature on using nutritional agents that raise Nrf2 during human pregnancies, agents that produce lowering defects in the fetus [278,279,280,281,282,283,284,285,286,287]. Ref. [285] may be especially relevant to ASD causation, because it shows that raising Nrf2 in the pregnant mother can prevent disease in the offspring. However, [278] expresses concern about raising Nrf2 to higher levels possibly producing problems during pregnancy via reductive stress. I have searched for additional literature on human or animal pregnancy and reductive stress and the only additional paper I have found is a paper on reductive stress producing lowered fetal attachment—a time period very distant from the perinatal period, which is thought to be central to ASD causation.

Agents that raise AMPK may be useful as well, although those agents that raise AMPK via lowered energy metabolism should be used with caution, especially if they are to be used during pregnancy.

Other approaches to treatment include all of the things mentioned above for prevention, but also using approaches such as putting the person with an ASD in an enriched environment, or using cognitive behavioral therapy, both of which are reported to produce clinical improvement in humans. The Hong et al., 2020 [133] study of in utero cell phone radiation exposure to induce a long-lasting, possibly ASD-like condition in rats. These same ASD aged rats were still suffering from learning disability, as well as lowered expression of three genes needed for proper synaptic structure, and aberrant synaptic properties in the rat hippocampus. Putting those aged rats in an environmentally-enriched living space allowed them to recover much learning ability, while improving synapse properties and increasing expression of the three genes involved in determining synaptic structure. The improvements in learning are associated with large improvements in synaptic structure and function. This may not be the only approach toward retraining and restructuring synapses in ASDs.

I have not said anything about avoiding chemical exposures. The best way to avoid the pesticides and glyphosate is to eat organic (in the US) or bio (in Europe) foods. Aluminum hydroxide or salts should not be eaten or injected. Aluminum in foods is often very poorly absorbed, although eating Al^3+^ ions with a citrate or other chelators can increase absorption by as much as 100-fold. Tea, which is fairly rich in aluminum, should not be drunk with lemon juice. Baking powder almost always has aluminum in it. The usually poor absorption of aluminum in foods means that comparing these with injected aluminum hydroxide or aluminum salts can be very misleading. Environmental exposures to PCBs and PFAS are difficult for most people to assess, although making sure people have a very good source of drinking water may be essential. Because raising Nrf2 increases detoxification of the body from both carbon-containing toxicants and toxic metals, agents that raise Nrf2 should always be considered when one is trying to minimize chemical causation of ASDs, or obtain therapeutic improvement of individuals with ASDs. People have become more aware of phthalate toxicity, allowing us to avoid some common phthalate sources. Mercury and mercurials should be avoided wherever possible.

This discussion of ASD treatment and prevention is not intended to be a complete discussion, but rather as a start, to initiate a wider discussion.

## Figures and Tables

**Figure 1 brainsci-14-00454-f001:**
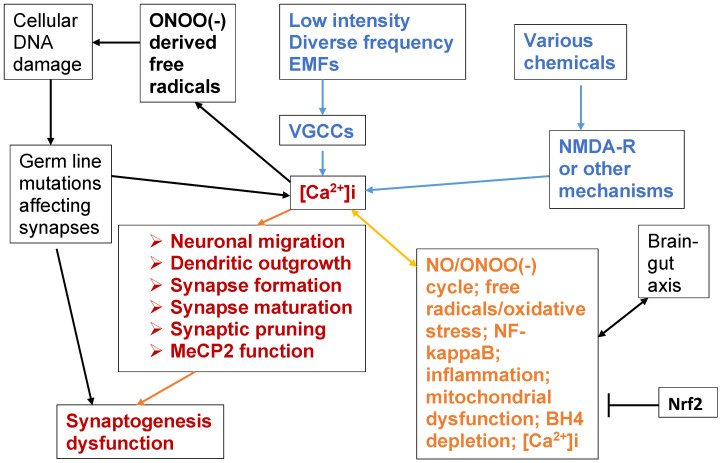
Proposed central set of mechanisms causing autism spectrum disorders by acting during the perinatal period (before and after birth), to impact synaptogenesis which normally occurs at very high rates. The most central mechanism is excessive intracellular calcium [Ca^2+^]i which causes abnormal regulation of each of the six mechanisms just below center left, each of which are essential to synaptogenesis. Each of those six mechanisms are regulated by [Ca^2+^]i. Electronically-generated EMFs activate the VGCC calcium channels in the plasma membrane, producing large calcium influxes and therefore large increases in [Ca^2+^]i. Each of the 15 chemicals, or groups of chemicals, that are thought to have roles in ASD causation also act to increase [Ca^2+^]i. The NO/ONOO(-) cycle mechanism, on the lower right side, is proposed to have a major role in producing the chronicity of ASDs.

**Figure 2 brainsci-14-00454-f002:**
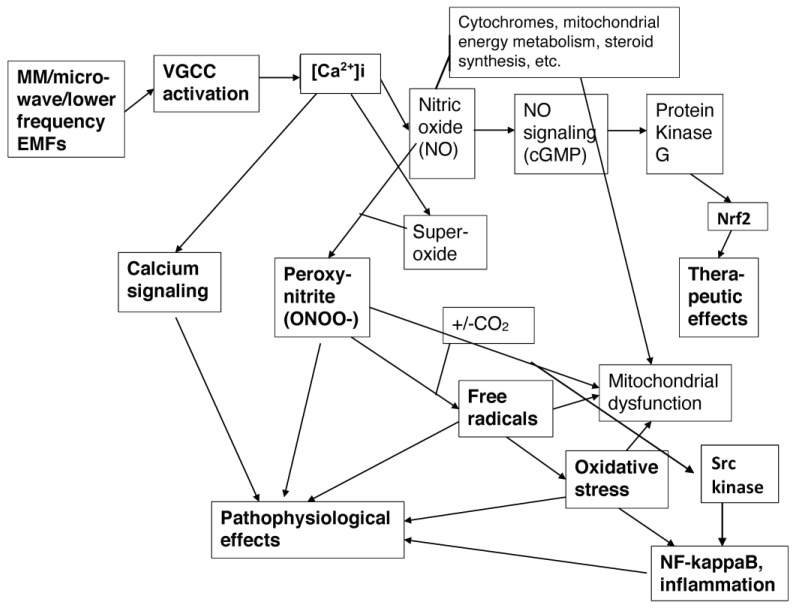
Taken with some modification from [91] with permission. These four pathways produce both pathophysiological effects and, in the case of one pathway, therapeutic effects. The main effects are produced via elevated calcium signaling, such as those discussed in Figure 1, which disrupts normal synaptogenesis. The peroxynitrite pathway going to the right can cause large elevations of peroxynitrite (ONOO- or ONOO(-)), a potent oxidant which breaks down to release highly reactive free radicals which produce oxidative stress. Peroxynitrite also produces large increases in NF-kappaB and inflammatory cytokines and inflammation. Oxidative stress and inflammation occur in almost all human chronic diseases, including ASDs. The therapeutic pathway occurs when there are only modest increases in [Ca^2+^]i. It produces effects that are almost opposite of the effects produced by the peroxynitrite pathway—these therapeutic effects are mostly produced by raising the levels of Nrf2.

**Figure 3 brainsci-14-00454-f003:**
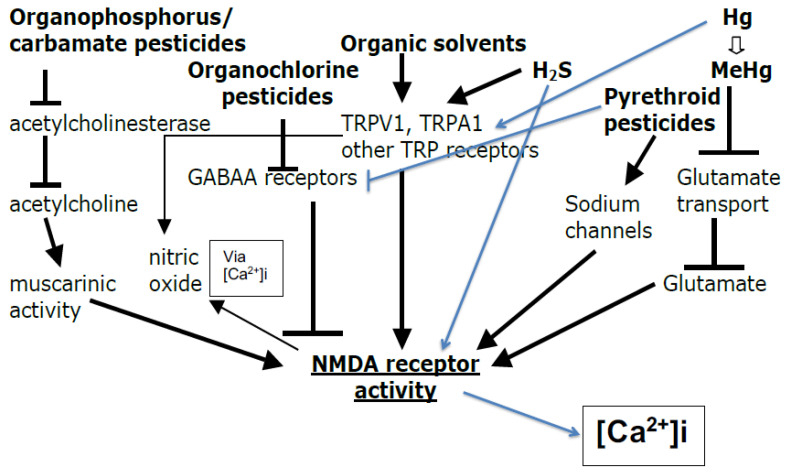
Pathways of action whereby mercury/mercurials, diverse organic solvents, and four classes of pesticides can each act to produce increased NMDA-R activity and consequently, increased [Ca^2+^]i. (Modified from [165] with permission).

**Figure 4 brainsci-14-00454-f004:**
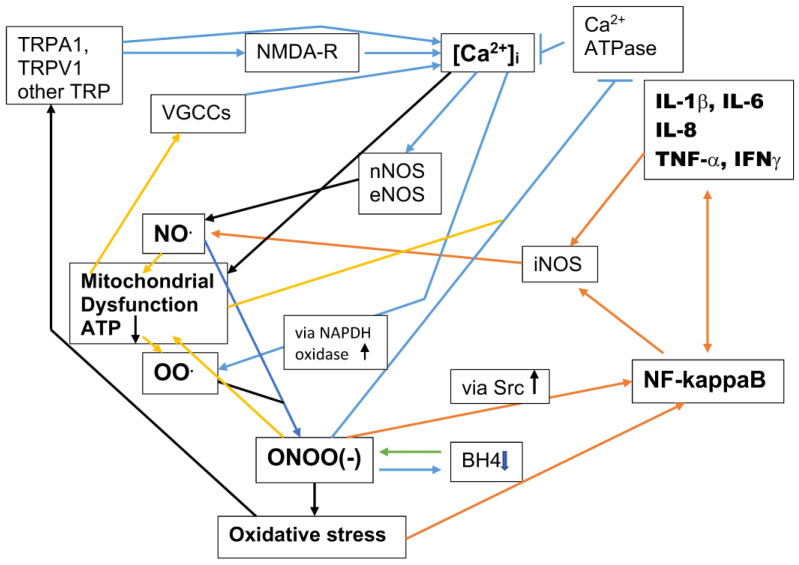
This is a diagram of the NO/ONOO(-) cycle where peroxynitrite and many other components form a vicious cycle, where everything is both a cause and an effect of the cycle. The five main centers of the NO/ONOO(-) cycle are discussed in the text. The cycle is most clearly described in [209], where the 34 distinct mechanisms making up the cycle are also each described. Because each of these 34 mechanisms are local, the cycle can impact various tissues in the body, producing diverse chronic diseases. Ten different elements in the cycle, as shown in Table 1, are elevated in patients with ASDs. Consequently, the NO/ONOO(-) is highly relevant to autism/ASDs. Arrows next to words are used to indicate either increased or decreased activity.

**Figure 5 brainsci-14-00454-f005:**
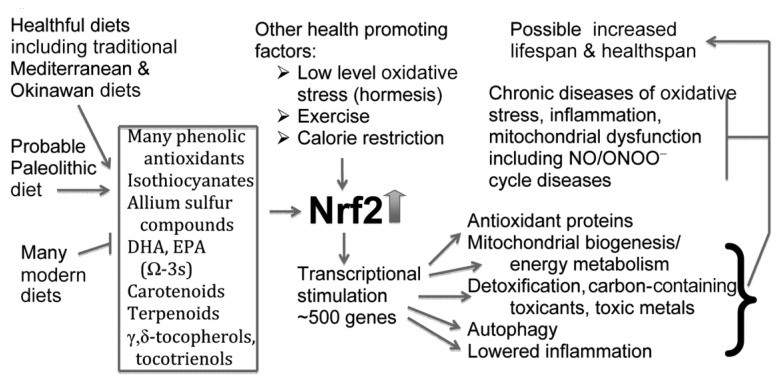
Raising Nrf2 produces many heath-promoting effects. While Nrf2 is most known for producing large increases in antioxidant enzymes, Nrf2 also raises other health-promoting effects, as shown in the lower right. It has the ability to lower NF-kappa and inflammation, to improve mitochondrial function. It acts to get rid of dysfunctional organelles and of destructive protein aggregates (via autophagy). It detoxifies the body of carbon-containing toxicants and toxic metals. These various effects help down-regulate the NO/ONOO(-) cycle. Findings are described in the text that strongly suggest that raising Nrf2 may help in the treatment of ASDs and may even ameliorate ASD cases by acting during pregnancy. Various nutrients that raise Nrf2 are discussed in [244].

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
