# Peer review of "Central Causation of Autism/ASDs via Excessive [Ca2+]i Impacting Six Mechanisms Controlling Synaptogenesis during the Perinatal Period: The Role of Electromagnetic Fields and Chemicals and the NO/ONOO(-) Cycle, as Well as Specific Mutations"

_brainsci, 2024, doi:10.3390/brainsci14050454_

Round 1

Reviewer 1 Report

Comments and Suggestions for Authors

I have reviewed the paper entitled "Central causation of Autism/ASDs via excessive [Ca2+]i impacting six mechanisms controlling synaptogenesis during the perinatal period. Role of EMFs and chemicals and the NO/ONOO(-) cycle as well as specific mutations." by Martin Pall.

While the topic is intriguing, there are significant improvements needed.

I suggest not to use the acronym EMFs in the title.

The introduction lacks clarity in its purpose and fails to adequately introduce the reader to the topic of autism and neurodevelopment. Instead of starting this section with “figure 1”, I suggest that the author should provide sufficient information on the topic so as to conclude with figure 1. In addition, figure 2 is presented in the manuscript before figure 1.

What does the author mean by “The discussion is presented in the introduction without documentation” in line 29 of the introduction…?

In regards to the "ASD epidemic," the author should provide epidemiological data to support this assessment and give weight to the argument.

The author should provide more detailed information regarding the significance of the biological signaling pathways mentioned in the manuscript. For example, the author refers to several biochemical steps that lead to Nrf2 activation in the first sections of the paper. However, it is not until the later sections that the author provides more information on the biological significance of Nrf2 activation. The same happens with the EPHA5 signaling (line 311 page 8). More information should be provided earlier in the text describing the biological significance of activating or inhibiting these signaling pathways.

This is essential for a broader audience, as it will allow readers without extensive biochemistry knowledge to better understand the mechanisms discussed in the paper.

 Providing clear explanations can significantly enhance the accessibility and relevance of the research findings to a wider range of readers.

Furthermore, the manuscript appears overly long and repetitive, which detracts from its impact.

One area that requires attention is the need for more transparency regarding the total number of papers and results from clinical studies suggesting a causative link between electromagnetic fields and ASD. This transparency is crucial to avoid bias and ensure a more comprehensive understanding of the research findings.

Overall, while the paper addresses an important topic, several key areas need refinement before it can be considered for publication in Brain sciences.

Reviewer 2 Report

Comments and Suggestions for Authors

The submitted review suggests a new look at the molecular mechanisms underlying ASDs development and provides recommendations for decreasing the risk of their occurrence. I have found this review interesting, but one major issue has to be addressed before publication. The possible role of glial cells is ignored in the review, but these cells are considered key players in synaptogenesis. In the section regarding NMDA receptors, the action of chemicals on these glutamate receptors is described in general. However, it is known that NMDARs are also expressed in astrocytes (please, see the works of Verkhratsky and coauthors for details). Therefore, excessive activation of astrocytic NMDARs should also increase [Ca2+]i in these glial cells. In this regard, it would be interesting to see in the text some information about the possible role of altered astrocytic Ca2+ signaling in ASDs development. Moreover, it should be noted that cation channels of the TRP family are also widely expressed in astrocytes, and, as some studies have demonstrated, these channels can be modulated by external EMF. 

Comments on the Quality of English Language

Typos correction is required. 

Reviewer 3 Report

Comments and Suggestions for Authors

The author's evident dedication to addressing public health concerns, particularly the intricate disorder of ASD, is commendable. Author has undertaken thorough literature research and reviewed previously reported underlying mechanisms, enriching this manuscript with a wealth of information for readers interested in ASD mechanisms.

However, while the manuscript effectively compiles a vast amount of data, it falls short in offering novel insights and clear future directions, which are typically expected in such comprehensive review articles. The burgeoning prevalence of ASD is undoubtedly a timely and critical topic, but it must be acknowledged that the youth population has increased significantly in recent decades, leading to greater access to diagnostic services. Thus, partially, accounting for rising number of ASD cases. That said, the exposure to the agents/factors shown to have some form of causal relation to ASD, has definitely increased and cannot be ignored as contributing element.

Although exposure to factors like EMFs and pesticides has risen and it’s certainly a matter to concern, without significant evidence of a causal relationship, advocating for the cessation of technologies such as Wifi, Bluetooth, and Cellphones, as suggested by the author, is impractical and irrational. The immense benefits derived from these technologies cannot be dismissed based on statistically unproven connections. Therefore, the discussion section, particularly the author's stance on these matters, needs to be reframed to adopt a cautious rather than fear-inducing tone.

The manuscript fails to report or discuss aspects like at what level of intracellular calcium levels ASD development becomes critical, and lacks suggestions for specific research directions on how to target this aspect.

Animal model based ASD studies, while promising and needed, do not necessarily translate directly to human cases, a fact that the author knows and despite that appears to be advocating for direct use of many of these agents.

Additionally, suggestions such as the use of agents like resveratrol seem highly risky, especially considering its known teratogenic effects.

The author has endeavored to elucidate the role of VGCC-induced increased calcium levels, but appears to downplay studies that demonstrate decreased VGCC activity, by using terms like "puzzling." While it's appreciated that the author acknowledges contrasting research, there is a lack of elaboration on these points to maintain the narrative flow.

In line 1064, the author need not apologize for omitting certain studies from the manuscript. Instead, focusing on fewer but more impactful citations would enhance the discussion's depth.

Similarly, instances where the context is vague or repetitive, such as in the paragraph on page 19, line 784, are concerning as they detract from the manuscript's clarity and coherence.

Minor errors like missing reference for citation Porokhovnik et al. [248] at line 1015, duplication of the reference 247 and 248, and figure 2 might be missing Nrf2.

Overall, the manuscript's extensive scope may be of limited benefit as it fails to provide clear and concise guidance to readers. The tone of the article tends towards casual rather than formal, resembling an opinion piece for a news outlet despite its comprehensive information. Thoughtful revision is needed, including a shift to a neutral and formal tone, and a reassessment of the discussion section to avoid unsubstantiated medical suggestions. In its current form, the latter half of discussion section may be better suited for publication in a public health opinion journal.

Round 2

Reviewer 2 Report

Comments and Suggestions for Authors

I think my comment has been addressed. 

Author Response

Thank you very much

Reviewer 3 Report

Comments and Suggestions for Authors

I appreciate the author for meticulous revision based upon the editor's and reviewers' feedback. Although, the revised version of the manuscript has undergone substantial improvements, I still have a few critical concerns.

The revised content in the discussion between 1113 and 1146, and particularly at the lines 1131,1136 and 1146, was what I meant to highlight as the issue of insufficient novelty. While the author’s commendable work demonstrates the convergence of multiple theoretical pathways, their diverse nature poses challenges in treating them as a unifying hypothesis. Accurate follow-up in future research becomes intricate due to this diversity. Additionally, I commend the author for including thoughtful sentences at lines 84-86.

The author’s statements in response to questions about animal models and resveratrol appear contradictory. It’s important to recognize that the scientific community engages in extensive debate and skepticism regarding the use of animal models for complex human behavioral disorders. The author himself acknowledges this at line 84. Consequently, it would be prudent for the author to refrain from offering medical advice related to the use of potential agents or natural products. This caution is especially relevant given the limited number of rodent-only studies and the manuscript’s scientific focus. I also raised this concern to prevent any dangerous misinterpretation of the statement at line 1035 by the general public or the media. Recent evidence during the pandemic highlights instances where individuals attempted self-medication without proper medical guidance. Terms such as “ideal” and “agents” in lines 1033-1036, particularly when directly referring to mothers, may falsely imply medical advice—an area where neither the author nor the manuscript is qualified to provide guidance. Therefore, I strongly recommend that the author rephrases lines 1034-1036, particularly considering the potentially carcinogenic nature of resveratrol.

Similarly, I would appreciate using a more cautionary word rather a fear generating tone e.g., at line 769, which can be easily picked and warped by the media.

The studies suggesting a connection between EMF and ASD lack compelling evidence and broad consensus. Notably, none of the referenced studies supporting the author’s hypothesis originate from reputable laboratories or institutes (e.g., references in sections 2.4, 2.5, and 6.1). It appears unlikely that the entire scientific community, including prominent labs and institutes worldwide, would overlook such a significant issue without questioning industry standards or government regulations. However, irrespective of my perspective, the cited studies remain limited in number, and even collectively, they necessitate further follow-up research to validate the hypothesis and achieve consensus.

Moreover, I respectfully disagree and strongly advise removing all content between lines 1160-1181 and 1205-1211. Most of these statements seem to reflect the author's personal beliefs without any clear scientific backing (no references provided by author). It's important to note that this is a scientific review, not an opinion piece where author can offer medical or social advice. Such recommendations could potentially harm pregnant women and their families and lead to financial consequences. Additionally, there is no evidence suggesting any conflict of interest with the LessEMF company. This company lacks proper accreditation, lacks scientific validation for the measures it sells, and is not endorsed by any government or scientific agency. Providing advice based on such unverified sources risks significant financial harm to readers, especially if sensationalized by news outlets.

Casual sentences must be revised, like one at the line 1164, “Yes I know…’ which could be interpreted as derogatory towards pregnant women, many of whom work diligently in urban areas and elsewhere to improve their lives and support their families. Based upon a few unvalidated rodent studies and author’s advice they cannot afford to leave the city and live in jungle or spend a fortune to attempt to shield themselves with unproven measures. Please remove it.

Similarly, author references rodent studies indicating benefits from elevated Nrf2 levels yet fails to cite any study which can serve as guide for sources to increase Nrf2 apart from acupuncture, which is considered an alternative therapy and lacks widespread medical acceptance. Additionally, the suggested sources in Figure 5 appear to be general health-promoting diets without scientific evidence specifically linking them to Nrf2 elevation, as no citations are provided by the author. Moreover, if Mediterranean and Okinawan diets are indeed effective, why have they not been successful in mitigating autism prevalence in those regions?

While the manuscript does provide valuable insights into the complexity of ASD etiology, it deviates from its central premise, as proposed in the manuscript tittle and abstract. In the end, the discussion section predominantly mirrors the author’s beliefs and primary objective. The focus is on electromagnetic fields (EMFs) and hypotheses related to chemical exposure, supported by a curated selection of biased references. Collectively, this gives the impression that the author aims to communicate this issue to a general audience while substantiating it with scientific evidence. If the target audience is the scientific community, the author should have elaborated more on the future research directions. Similarly, the author should have prioritized including studies or sources that provide additional information on safety guidelines for EMFs. Instead, the manuscript predominantly attributes unproven facts to industry and agencies.

The overall lack of compelling support for many of the expansive claims implies an attempt to validate opinions rather than establish facts through rigorous scientific methods. Further revision is necessary to address these issues, particularly by removing sections where the author offers unproven advice in the discussion section.
